# A high-resolution satellite-based map of global methane emissions reveals missing wetland, fossil fuel and monsoon sources

Xueying Yu[1,2], Dylan B. Millet[1*], Daven K. Henze[3], Alexander J. Turner[4], Alba Lorente Delgado[5], A. Anthony Bloom[6], Jianxiong Sheng[7]

[1]Department of Soil, Water, and Climate, University of Minnesota, Saint Paul, Minnesota 55108, United States
[2]Department of Earth System Science, Stanford University, Palo Alto, California 94305, United States
[3]Department of Mechanical Engineering, University of Colorado, Boulder, Colorado 80309, United States
[4]Department of Atmospheric Sciences, University of Washington, Seattle, Washington 98195, United States
[5]Earth Science Group, SRON Netherlands Institute for Space Research, Leiden, the Netherlands
[6]Jet Propulsion Laboratory, California Institute of Technology, Pasadena, California 91109, United States
[7]Center for Global Change Science, Massachusetts Institute of Technology, Cambridge, Massachusetts 02139, United States

*Correspondence to*: Dylan B. Millet (dbm@umn.edu)

**Abstract.** We interpret space-borne observations from the TROPOspheric Monitoring Instrument (TROPOMI) in a multi-inversion framework to characterize the 2018–2019 global methane budget. Evaluation of the inverse solutions indicates that simultaneous source + sink optimization using methane observations alone remains an ill-posed problem—even with the dense TROPOMI sampling coverage. Employing remote carbon monoxide (CO) and hydroxyl radical (OH) observations with independent methane measurements to distinguish between candidate solutions, we infer from TROPOMI a global methane source of 587 (586–589) Tg/y and sink of 571 Tg/y for our analysis period. We apply a new downscaling method to map the derived monthly emissions to 0.1°×0.1° resolution, using the results to uncover key gaps in the prior methane budget. The TROPOMI data point to an underestimate of tropical wetland emissions (a posteriori increase of +13 [6–25] % or 20 [7–25] Tg/y), with adjustments following regional hydrology. Some simple wetland parameterizations represent these patterns as accurately as more sophisticated process-based models. Emissions from fossil fuel activities are strongly underestimated over the Middle East (+5 [2–6] Tg/y a posteriori increase) and over Venezuela. The TROPOMI observations also reveal many fossil fuel emission hotspots missing from the prior inventory, including over Mexico, Oman, Yemen, Turkmenistan, Iran, Iraq, Libya, and Algeria. Agricultural methane sources are underestimated in India, Brazil, the California Central Valley, and Asia. Overall, anthropogenic sources worldwide are increased by +19 [11–31] Tg/y over the prior estimate. More than 45% of this adjustment occurs over India and southeast Asia during the summer monsoon (+8.5 [3.1–10.7] Tg in Jul–Oct), likely due to rainfall-enhanced emissions from rice, manure, and landfills/sewers, which increase during this season along with the natural wetland source.

**Short Summary.** We combine satellite measurements with a novel downscaling method to map global methane emissions at 0.1°×0.1° resolution. These fine-scale emission estimates reveal unreported emission hotspots and shed light on the roles of agriculture, wetlands, and fossil fuels for regional methane budgets. The satellite-derived emissions point in particular to

missing fossil fuel emissions in the Middle East and to a large emission underestimate in South Asia that appears to be tied to monsoon rainfall.

## 1 Introduction

Methane ($CH_4$) has a 20-year global warming potential 85 times that of carbon dioxide ($CO_2$) and is an important driver of decadal climate changes (IPCC, 2021). Global mean methane mole fractions reached 1879 ppb in 2020, 2.6× pre-industrial levels, with a recent growth rate acceleration (+10–15 ppb/y in 2019–2020) whose cause is not well understood (NOAA, 2022; Peng et al., 2022; Saunois et al., 2020; Stevenson et al., 2021). Strong spatial and temporal heterogeneity in methane emissions and limited observational coverage have historically challenged our ability to explain such trends in terms of underlying sources. However, the recent availability of high-resolution, near-global and daily methane measurements from the TROPOspheric Monitoring Instrument (TROPOMI) provides a transformative advance in this area. Here, we apply these data in a 4D-Var inversion + spatial downscaling framework to quantify the 2018–2019 global methane budget and determine the importance of missing and unexpected sources.

The atmospheric methane burden increased by an average of 18 (17–19) Tg/y during 2008–2017 (Saunois et al., 2020), with conflicting explanations proposed. Top-down studies have inferred a ~30 Tg/y emission increase over tropical regions between 2000–2006 and 2017 (Bergamaschi et al., 2013; Lunt et al., 2019; Jackson et al., 2020). However, such inferences can be highly sensitive to even modest uncertainties in the atmospheric hydroxyl radical (OH, the main methane sink)—particularly over the tropics with their sparse observations (Mcnorton et al., 2016; Rigby et al., 2017; Turner et al., 2017). Some top-down studies have approached this problem by co-optimizing methane emissions and sinks (Lu et al., 2021; Maasakkers et al., 2019; Qu et al., 2021; Turner et al., 2017; Zhang et al., 2018). Unfortunately, these terms may be insufficiently resolved for robust inverse analysis when using methane data alone, leading to aliasing between the optimized source and sink terms (Lu et al., 2021; Zhang et al., 2021). Isotopic analyses invoke increased biogenic sources (Nisbet et al., 2016; Schaefer et al., 2016) to explain the post-2016 $^{13}C$ depletion, whereas ethane-based constraints indicate a fossil fuel emission underestimate (Franco et al., 2016; Kort et al., 2016; Peischl et al., 2016; Xiao et al., 2008). Unfortunately, the latter approach is limited by the large variability in methane-to-ethane emission ratios.

Bottom-up inventories also point to substantial uncertainties in the spatial distribution of methane sources. For instance, the two most commonly-used anthropogenic inventories for the US (EDGAR v5 (2019) and GEPA (Maasakkers et al., 2016)) are essentially uncorrelated (R = 0.1) at 0.1°×0.1° resolution. Meanwhile, current inventories also lack the ability to predict emission sporadicity (e.g., Irakulis-Loitxate et al., 2022; Pandey et al., 2019), while temporal representation errors can also arise between inventories due to time lags associated with their development. Such biases, when coupled with sparse observations, model transport errors, and source/sink ambiguity, degrade the accuracy of observation-based (top-down)

emission estimates—which as a consequence often arrive at inconsistent emission allocations (Alexe et al., 2015; Bruhwiler et al., 2014; Jackson et al., 2020; Lu et al., 2021; Maasakkers et al., 2019; Mcnorton et al., 2018; Monteil et al., 2013; Qu et al., 2021; Yu et al., 2020; Yu et al., 2021a; Yu et al., 2021c; Zhang et al., 2021).

TROPOMI provides an unprecedented observational expansion for addressing these science gaps, offering sub-10 km global monitoring of total column methane concentrations with dense overland coverage (Bousserez et al., 2016; Jacob et al., 2016; Maasakkers et al., 2022; Turner et al., 2018b). Here, we interpret two years of TROPOMI data in an analysis framework that couples multiple 4D-Var adjoint inversions with a novel spatial downscaling approach to derive emissions at $0.1° \times 0.1°$

horizontal resolution. This yields a suite of candidate solutions for the 2018-2019 methane budget, which we evaluate a posteriori against independent observations of methane, carbon monoxide (CO), and OH. In this way we identify the most robust solution set based on the ensemble of observational constraints, and use this new spatial information to better understand regional and sectoral contributions to the methane budget and the underlying drivers of those emissions.

## 2 Data and Methods

Figure 1 summarizes our inversion framework. We employ TROPOMI measurements from 01/2018–02/2020 with the GEOS-Chem adjoint model in a suite of 4D-Var inversions to optimize monthly total methane emissions at $2° \times 2.5°$ (latitude × longitude) resolution. These derived emissions are then spatially downscaled to $0.1° \times 0.1°$. We omit the first and final 4 months from interpretation to further minimize initial condition errors and to ensure that all derived fluxes are adequately informed by subsequent observations. Our final analysis timeframe thus spans 18 months from 05/2018 through

10/2019.

### 2.1 TROPOMI observations and independent evaluation datasets

TROPOMI was launched in 10/2017 onboard the Copernicus Sentinel-5 Precursor satellite into a low-Earth polar orbit, and monitors greenhouse gases and air pollutants with daily near-global coverage at ~13:30 LT (equator overpass) on the ascending node (Hu et al., 2018). We use the SRON corrected retrieval described in Lorente et al. (2021), which is based on

the S5P-RemoTeC full-physics algorithm with albedo correction and updated regularization scheme, spectroscopic information, and surface treatment. This updated algorithm mitigates the albedo bias that affected earlier versions (Qu et al., 2021). Relative to the albedo-corrected product, the prior TROPOMI version exhibits high biases over North Africa, the Middle East, and the western US, and low biases over Amazonia, the eastern US, central Africa, and eastern China (Lorente et al., 2021).

The TROPOMI total column ($XCH_4$ in ppb) retrievals employ combined solar backscatter measurements in the near-infrared (NIR; 0.8 μm) and shortwave-infrared (SWIR; 2.3 μm), and have $5.5/7 \times 7$ km$^2$ nadir resolution on a 2600 km swath. The

data have <1% nominal bias, 0.6% instrument noise, and an estimated 0.8% forward model error (Hu et al., 2016). We omit high-latitude (>60°) observations and require quality filter QA > 0.5 (Sentinel-5 Precursor/TROPOMI Level 2 Product User Manual: Methane, 2022) to avoid errors associated with high solar or viewing zenith angles, low surface albedo, excessive aerosol loading, clouds, terrain roughness, and measurement noise (Lorente et al., 2021). Figure 2 shows the resulting TROPOMI XCH$_4$ data for 03/2018–02/2020, gridded to 0.1°×0.1° using the method described by Sun et al. (2018). In total, 91 million retrievals during 05/2018–10/2019 pass quality filtering and are available for analysis, an average of 31,000 per 2°×2.5° GEOS-Chem grid cell (Figure S1). For inversions on the 2° × 2.5° model grid, we first average the TROPOMI observations to this resolution.

Figure S2 shows that TROPOMI measurements within our inversion timeframe agree well with independent measurements from the Total Carbon Column Observing Network (TCCON; 2014) and the Greenhouse Gases Observing Satellite (GOSAT; 2021), with major axis regression slopes of 1.02 (R = 0.82) and 0.99 (R = 0.88), respectively. The inter-dataset mean biases are -7.1 ppb (0.4%, TROPOMI - GOSAT) and -5.4 ppb (0.3%, TROPOMI - TCCON; see Text S1). Our initial condition optimization further ensures that the model and TROPOMI are unbiased with respect to each other, so that mismatches arising during the simulation timeframe reflect source-sink disparities rather than any systematic observational bias.

We use a large suite of independent measurements to evaluate the inversions. These include methane columns from the TCCON (2014) network of Fourier transform spectrometers, and methane mole fractions from the ObsPack (near-real time version v2.0; (2021)) compilation of ground-based and airborne measurements. We further use CO and OH measurements from the Atmospheric Tomography (ATom) airborne campaign (Wofsy et al., 2018) to test inversion success at separately optimizing methane sources and sinks. ATom featured pole-to-pole profiling (0.2 to 12 km) during four seasons over four years. The flight design is thus well-suited to determine whether the optimized OH fields improve or degrade global model simulations of OH itself and of CO (whose dominant sink is reaction with OH). Measurements of CO during ATom were performed using the NOAA Picarro instrument with an estimated uncertainty of 3.6 ppb (Chen et al., 2013). OH measurements during ATom employed the Airborne Tropospheric Hydrogen Oxides Sensor (ATHOS), with an estimated uncertainty of 0.018 ppt (1-minute average; Brune et al., 2020).

**2.2 Forward model and initial conditions**

We use the GEOS-Chem adjoint model (v35), on a 2° × 2.5° grid with 47 vertical layers, to perform the global 4D-Var inversions. The model uses GEOS-FP meteorological fields from the National Aeronautics and Space Administration (NASA) Global Modeling and Assimilation Office (GMAO, 2013), with 5- and 10-minute timesteps for transport and emissions, respectively. Transport employs fully instantaneous boundary layer mixing (Wu et al., 2007), a relaxed Arakawa-Schubert convection scheme (Moorthi and Suarez, 1992), and a multi-dimensional Flux-Form Semi-Lagrangian (FFSL)

treatment for advection (Lin and Rood, 1996). For all model-satellite comparisons (and at each inversion iteration) the GEOS-Chem output is sampled according to the TROPOMI observation operator at the overpass time and location.

We optimize the model initial conditions for 01/01/2018 in three steps, first starting with a 25-year global spin-up to achieve a globally representative methane field. We then apply a latitude-dependent correction based on the TROPOMI-model difference for 11/2017–01/2018 to speed up the optimization process in the next step. Correction over land employs the median TROPOMI-model difference by latitude; over oceans (which lack TROPOMI $XCH_4$ data) we use the 0.1 quantile difference. Finally, we optimize the resulting fields in a 4D-Var inversion based on TROPOMI data for 01–02/2018. The optimized global methane burden is 0.99× that in the original 25-year spinup, and the north:south hemispheric (NH:SH) $XCH_4$ ratio increases from 1.11 to 1.13.

## 2.3 Prior model sources and sinks

Global anthropogenic methane emissions in the prior model use the gridded United Nations Framework Convention on Climate Change (UNFCCC) inventory for fossil fuels (year-2016; GFEI (Scarpelli et al., 2020)) and EDGAR v5 for other sources (year-2015; (Crippa et al., 2019; Crippa et al., 2020)). These are superseded by the 2012 GEPA inventory for US anthropogenic emissions (Maasakkers et al., 2016), the CanMex inventory for Canadian (year-2013) and Mexican (year-2010) oil and gas emissions (Sheng et al., 2017), and Sheng et al. (2019) for Chinese coal mine emissions (year-2011). Wetland emissions use the 2018–2019 WetCHARTs ensemble mean flux (Text S2; (Bloom et al., 2017)), scaling the total to 149 Tg/y to match the 2008–2017 global methane budget from Saunois et al. (2020). We apply Fung et al. (1991) and Maasakkers et al. (2019) for termite and geological seep emissions, respectively, and employ biomass burning emissions for 2018–2019 from the Quick Fire Emissions Dataset (QFED (Darmenov and Silva, 2015; Koster et al., 2015)). Figure S3 maps these prior emissions, which total 535 Tg/y and include 356 Tg/y from anthropogenic sources (119 Tg/y livestock + 101 Tg/y fossil fuel + 80 Tg/y waste + 37 Tg/y rice + 19 Tg/y other), 165 Tg/y from natural sources (149 Tg/y wetlands + 16 Tg/y geological seeps and termites), and 14 Tg/y from biomass burning. Emissions from EDGAR v5 vary monthly based on national and sub-national sectoral activity levels. GEPA includes monthly emission profiles for US rice and manure management, and we assume aseasonal emissions for indoor animal husbandry following Crippa et al. (2020). The WetCHARTs emissions are monthly, reflecting temporal changes in wetland extent, respiration, and temperature. QFED emissions are daily with an hourly diel profile applied.

Atmospheric methane removal by OH (87% of the total sink, 494 Tg/y) in the prior model uses archived monthly oxidant fields from GEOS-Chem (v5) benchmark simulations (Wecht et al., 2014), which have an annual tropospheric air-mass-weighted mean of $1.03 \times 10^6$ molecules/cm$^3$ and 1.04 NH:SH ratio. Other minor sinks include stratospheric oxidation (6%, 33 Tg/y) based on NASA Global Modeling Initiative monthly loss frequencies (Murray et al., 2013), soil uptake (6%, 34 Tg/y) following Fung et al. (1991), and tropospheric oxidation by chlorine (2%, 10 Tg/y) using 3-D monthly Cl fields from

Sherwen et al. (2016). The above sinks total 571 Tg/y (Figure 2e) and yield a 9.1-year methane lifetime in our prior
simulations.

Simulations to evaluate posterior model performance for CO and OH employ anthropogenic emissions (for CO, $NO_x$, and
VOCs) from the Community Emissions Data System (Hoesly et al., 2018), the 2016 EPA NEI v1 (NEIC, 2019), and the Air
Pollutant Emission Inventory (APEI, 2020). Corresponding biogenic and biomass burning emissions are obtained from the
170 Model of Emissions of Gases and Aerosols from Nature (MEGANv2.1; Hu et al., 2015), and QFED (Koster et al., 2015).

## 2.4 Inversion frameworks and sensitivity to OH

Optimizations are performed in the GEOS-Chem adjoint model (Henze et al., 2007) through iterative minimization of the
Bayesian cost function $J(\boldsymbol{x})$:

$$J(\boldsymbol{x}) = (\boldsymbol{x} - \boldsymbol{x_a})^{\mathrm{T}} \mathbf{S}_a^{-1} (\boldsymbol{x} - \boldsymbol{x_a}) + \gamma (\boldsymbol{y} - F(\boldsymbol{x}))^{\mathrm{T}} \mathbf{S}_0^{-1} (\boldsymbol{y} - F(\boldsymbol{x})) \qquad (1)$$

The first right-hand term imposes a penalty based on the deviation of $\boldsymbol{x}$ (the state vector to be optimized) from $\boldsymbol{x_a}$ (the prior
estimates), weighted by the prior error matrix $\mathbf{S_a}$. The state vector $\boldsymbol{x}$ includes monthly $2° \times 2.5°$ grid-level emissions and (in
some cases) the 2-year-mean hemispheric loss to OH. This penalty is counteracted by the second right-hand term, which
reflects the mismatch between the observations $\boldsymbol{y}$ and model predictions $F(\boldsymbol{x})$ sampled in the same manner, weighted by the
observing system error matrix $\mathbf{S_o}$. The regulation parameter $\gamma$ is applied to balance the influence of the above two terms in
the overall cost function $J(\boldsymbol{x})$. Our inversions run continuously from 01/2018 to 02/2020, optimizing monthly grid-total
180 methane emissions and 26-month mean hemispheric OH concentrations. To minimize any effects from initial conditions and
to allow sufficient observational constraints throughout the analysis period we focus interpretation on the 18-month period
from 05/2018 to 10/2019. Annual values discussed later are for 11/2018–04/2019 plus the mean of 05–10/2018 and 05–
10/2019.

The prior error covariance matrix $\mathbf{S_a}$ for methane emissions is constructed as follows. We use the Maasakkers et al. (2016)
scale-dependent uncertainties for anthropogenic emissions over the GEPA and CanMex domains, and the Sheng et al. (2019)
province-level error estimates for Chinese coal mine emissions. For other global anthropogenic emissions we use the gridded
fossil fuel uncertainty estimates from Scarpelli et al. (2020) and assume 50% uncertainty in the remaining sources
(Maasakkers et al., 2019; Yu et al., 2021c). Uncertainties for wetland emissions are derived as one standard deviation across
the WetCHARTs ensemble, averaging 105% at the grid level. Other sources employ a prior error standard deviation of 50%,
consistent with earlier studies (Maasakkers et al., 2019; Sheng et al., 2018; Turner et al., 2015; Wecht et al., 2014; Yu et al.,
2021c). The diagonal of the prior error matrix combines the above flux-weighted terms in quadrature and averages 66%. We
find that the spatial covariance in the total prior emissions decreases by 50% over a mean distance of approximately 300 km,

and we populate the exponentially decaying off-diagonal elements of $\mathbf{S_a}$ accordingly. This is comparable to the 200–500 km correlation length scales applied in previous methane studies (Monteil et al., 2013; Wecht et al., 2014; Yu et al., 2021c).

We test the impacts of OH on our results through three separate inversion treatments: the first uses the prior OH with no optimization, while the second and third optimize methane loss to OH on a hemispheric basis with an assigned uncertainty (included in $\mathbf{S_a}$) of either 1% or 10% (Prather et al., 2012; Saunois et al., 2020). We find that both the 1% and 10% uncertainty assumptions for OH give effectively identical inversion solutions: in both cases the optimization has sufficient OH flexibility to correct a global mean budget imbalance on that basis, with the remaining spatial errors resolved through grid-level emission adjustments (Figure S4). We therefore mainly discuss results from the 1% optimization (referred to as optOH), along with those from the fixed OH (fixOH) inversion. Other minor sinks, such as soil uptake, are also uncertain but not addressed here.

Observing system errors combine measurement errors and forward model errors. Building on Heald et al. (2004), we compute the elements of $\mathbf{S_o}$ (which is diagonal) from the residual standard deviation between the observations and the prior simulations within a $2° \times 2°$ moving window, further imposing a lower bound of 56 ppb$^2$ (0.25 quantile of the overall results). The resulting observing system errors average 11 ppb, mainly reflecting instrument noise, and are comparable to previous estimates for GOSAT and TROPOMI (11–13 ppb; (Maasakkers et al., 2019; Zhang et al., 2020)).

The regulation parameter $\gamma$ is defined through sensitivity inversions for 01/2018 with $\gamma$ varying from $10^{-5}$ to $10^3$. The optimal monthly value is selected based on the resulting L-curve and total error reductions (Figure S5), and then scaled to the number of observations for the full 2-year inversion period. The result ($\gamma = 0.03$) is consistent with previous TROPOMI inversions by Qu et al. (2021) ($\gamma = 0.002$) given their optimization of annual rather than monthly emissions.

Adjoint 4D-Var inversions do not directly provide posterior error estimates. Methods are available to indirectly derive such estimates (Bousserez et al., 2015; Yu et al., 2021a); however, our previous Observing System Simulation Experiments (OSSEs; Yu et al., 2021a) showed that for methane the computed posterior error reductions do not correlate with more accurate flux estimates in the presence of model transport errors and spatially inaccurate prior emissions. Here, we instead combine multiple inversion strategies (Section 2.5) with an ensemble of independent observations (Sections 2.1, 3) to test the robustness of our results and characterize the associated uncertainties.

Recent inverse analyses by Qu et al. (2021) likewise examined the global methane budget using TROPOMI (and GOSAT) observations. Our study advances on that work in several ways. First, we optimize monthly rather than annual fluxes to identify seasonal patterns of variability. Second, in place of a traditional analytical optimization we combine 4D-Var with new inverse formalisms for better identification of missing sources. Third, we develop a new downscaling approach to

constrain emissions at high resolution, and use this framework to elucidate flux mechanisms and missing sources. Finally, our analysis leverages an updated TROPOMI methane product (Lorente et al., 2021) that corrects an albedo-dependent bias present in the version used by Qu et al. (2021).

## 2.5 Inversion ensemble to explore sensitivity to missing sources

Our previous OSSE-based work (Yu et al., 2021a) emphasized that classical SF-based inversions have limited ability to recover missing sources. Here, we apply multiple inversion formalisms to diagnose and address this issue.

1. Classical scaling factor (SF) inversions, employing the bottom-up inventories described earlier as prior. These inversions solve for scale factors $s$ in $x = s \circ x_a$.

2. Background increment inversions (BI). BI inversions employ a revised prior consisting of the above inventories scaled by 90% plus the remaining 10% as a uniform overland flux. This revised prior is then subjected to SF optimization.

3. Observational guess inversions (OG). OG inversions employ a revised prior informed by long-term TROPOMI data for better recovery of missing sources. Specifically, we find from sensitivity simulations that adding 75 Tg/y to the bottom-up emissions results in a globally unbiased simulation across 2018–2020. We distribute this 75 Tg/y spatially based on the observation-model enhancement mismatches, where the grid-level enhancements are computed as the local XCH$_4$ value minus the zonal mean (2° bins). Figure S6 shows the resulting grid-level emission increments.

4. Emission enhancement inversions (EE). EE inversions solve for absolute flux increments rather than scale factors via $x = s \circ x_{base} + x_a$. We define $x_{base}$ as 2600 kg/box/timestep, which is the mean emission for grid cells exceeding 1 kg/box/timestep. We showed previously that this approach has better performance for missing sources than the above three SF inversions (Yu et al., 2021a).

In the following, we interpret the multi-inversion mean as our base-case solution and the range as the corresponding uncertainty estimate.

## 2.6 Emission downscaling

We present here a new method to spatially downscale the satellite-derived emissions from 2° × 2.5° to 0.1° × 0.1° for potential use in models. The downscaling, which combines information from the TROPOMI column enhancements, the prior emission estimates, and their uncertainties, is necessitated by the fact that the current GEOS-Chem adjoint model does not have global simulation capability at finer than 2° × 2.5° resolution. Furthermore, each of the 2-year inversions performed here required >12,000 CPU hours (>80 days on multiple processors) to converge, making higher-resolution optimizations computationally impractical. However, the inventories employed as prior, as well as the TROPOMI observations themselves, contain information at much finer scales (e.g., 0.1° × 0.1° and 7 × 7 km$^2$)—and thus contain additional high-resolution constraints that are neglected by the 2° × 2.5° inversions. We therefore leverage this information to spatially downscale the optimized emissions to 0.1° × 0.1° via:

$$x_j' = \omega_i \beta_{OBS,i\to j} s_i x_{a,j}' + (1 - \omega_i)\left(1 + \beta_{prior,i\to j}(s_i - 1)\right)x_{a,j}' \tag{2}$$

Equation (2) downscales the original optimized emissions from a given $2° \times 2.5°$ parent grid cell $i$ to the subgrid scale ($x_j'$ at

$0.1° \times 0.1°$) by combining spatial information from the observations (first right-hand term) and the prior (second right-hand term). Here, $\omega_i$ is a weighting factor to balance these two terms, $\beta_{OBS,i\to j}$ and $\beta_{prior,i\to j}$ are spatial downscaling operators representing the observational and prior information, respectively, $x_{a,j}'$ represents the $0.1° \times 0.1°$ prior emissions for subgrid $j$, and $s_i$ is the $2° \times 2.5°$ scale factor derived for parent grid cell $i$.

The observational downscaling operator $\beta_{OBS,i\to j}$ spatially allocates the subgrid-level emissions according to the distribution of column enhancements over the regional background:

$$\beta_{OBS,i\to j} = (y_{2y,j} - y_{bg,i})\bigg/\sum_{k \in j}\left(f_k(y_{2y,k} - y_{bg,i})\right) \tag{3}$$

where $y_{2y,j}$ is the 2-year mean (03/2018–02/2020) TROPOMI methane column sampled to $0.1° \times 0.1°$ (Sun et al., 2018) and $y_{bg,i}$ is the methane background defined as 95% of the $y_{2y,j}$ 0.1 quantile across the parent $2° \times 2.5°$ grid cell $i$. This background definition was determined via OSSE analysis (described below), with the corresponding parameters varied

systematically over a wide range to identify values yielding the best consistency with the true underlying fine-scale emissions. $f_k$ quantifies the prior fraction of total $2° \times 2.5°$ emissions contained in each subgrid $k \in j$.

The prior downscaling operator $\beta_{prior,i\to j}$ spatially allocates the derived flux enhancement $s_i - 1$ based on the prior emission magnitudes and their uncertainties:

$$\beta_{prior,i\to j} = \varepsilon_{a,j}' f_j\bigg/\sum_{k \in j}\left(f_k^2 \varepsilon_{a,k}'\right) \tag{4}$$

where $\varepsilon_{a,j}'$ is the prior emission error estimate at $0.1° \times 0.1°$. In this way larger corrections are preferentially assigned to locations with higher prior emissions and uncertainties.

When summed to $2° \times 2.5°$, both the prior and observational downscaling terms maintain the original adjoint-derived emissions. Weighting between these terms is computed as:

$$\omega_i = \varepsilon_{a,i}\hat{\sigma}_{a,i}\big/\max_{l \in i}(\varepsilon_{a,l}\hat{\sigma}_{a,l}) \tag{5}$$

where $\varepsilon_{a,i}$ is the prior emission error estimate at $2° \times 2.5°$, and $\hat{\sigma}_{a,i}$ is the log-transformed standard deviation of all $0.1° \times 0.1°$ prior emissions contained in that parent grid cell (with an imposed zero lower bound). As shown in Figure S7, the resulting downscaling relies most frequently on the prior information, particularly for low-emission areas, but hotspots and locations with higher prior uncertainties are preferentially informed by the observations.

Compared to existing emission downscaling methods that rely on prior and posterior error covariance estimates (Cusworth et al., 2021), or are based solely on satellite data (Liu et al., 2021), our approach is unique in combining the prior emission information (and its uncertainty) with the oversampled TROPOMI observations themselves. Variable weighting between these terms permits greater influence from the observations when the prior emissions are more uncertain. The method thus assumes robust prior error estimates, a caveat that also applies to Cusworth (2021) and similar methods.


    We tested the effectiveness of this downscaling approach in a 1-month OSSE analysis over North America (which features all relevant source types and a computationally tractable model domain). These synthetic inversions follow Yu et al. (2021a) in prescribing true and prior emissions from distinct inventories (true: Gridded EPA + WetCHARTs ensemble mean; prior: EDGAR v5 + a single WetCHARTs member) that differ by 76 Gg/d domain-wide and have spatial R = 0.51 at 0.25° ×

0.3125°. The OSSEs were performed at 2° × 2.5° by sampling the true-state model according to the TROPOMI coverage for 08/2018 (with measurement noise applied), followed by 4D-Var optimization as detailed by Yu et al. (2021a). The 2° × 2.5° adjoint solution was then spatially downscaled to 0.25° × 0.3125° following Eq. (2) and compared to both the true fluxes and to the adjoint solution performed directly on the fine-scale grid. Tests were performed both in the presence and absence of model transport error (Yu et al., 2021a).


    Table 1 shows that in the absence of transport error our downscaling approach outperforms the coarse-grid solution and approaches the skill of the native fine-scale inversion in representing the true fluxes. The benefits of the 4D-Var + downscaling approach are even more pronounced when accounting for transport error. Specifically, our previous OSSE analyses showed that high-resolution 4D-Var inversions failed to improve methane emission estimates at 25 km for scenarios

with both transport error and spatially biased emissions (Yu et al., 2021a). Our tests here show that spatial downscaling of the 2° × 2.5° adjoint solution strongly mitigates these effects (Table 1), yielding a larger bias reduction (98% versus just 16%) and more accurate flux distribution (R = 0.70 versus 0.60) than the native fine-grid 4D-Var solution. Figure S8 further shows that the downscaled OSSE solution reduces the prior bias by 17%–56% for sources exceeding 1000 kg $CH_4$/box/day (accounting for 99% of the domain-wide emissions) when not subject to transport error. In the presence of transport error,

the downscaling method has limited improvement for the very largest sources ($>2\times10^5$ kg/box/day), but nevertheless exhibits strong bias reduction (21%–50%) for sources between $1\times10^3$–$2\times10^5$ kg/box/day (which account for 96% of domain-wide emissions). Given these results and the finer-scale information available here, for the present work we apply Eq. (2) to spatially downscale our inversion solutions to 0.1° × 0.1°.

# 3 Derived Global Methane Budget and Sensitivity to OH

## 3.1 Methane source-sink ambiguity

The set of inversion configurations includes multiple formalisms for emission adjustment and two separate treatments for methane loss: fixOH inversions use the prior OH as a fixed constraint, while optOH inversions optimize both OH concentrations (as a 2-year hemispheric mean) and methane emissions. Figure 2 shows that the fixOH and optOH multi-model means yield similar atmospheric methane distributions, with the strongest enhancements over central Africa, South and East Asia, the Middle East, Amazonia and the southern US, and the lowest column concentrations over high southern latitudes in Australia, South America, and Africa. The two approaches also provide comparable improvement with respect to the TROPOMI observations, with >97% mean bias reduction and >45% root-mean-square-error (RMSE) reduction.

Despite the above patterns of agreement, the fixOH and optOH inversions lead to opposing methane emission changes relative to the prior budget. The fixOH multi-model mean provides a global methane source of 587 Tg/y (10% higher than the prior) and a sink of 571 Tg/y. The optOH multi-model mean yields a 514 Tg/y source (4% below the prior) and a 492 Tg/y sink; the latter is driven by updated OH fields with an air-mass-weighted NH:SH ratio of 0.98. Figure 2e shows that the fixOH and optOH solution sets adhere closely to a linear relationship between global sources and sinks, in all cases with a ~20 Tg/y growth rate in the atmospheric methane burden.

While mutually inconsistent, the fixOH and optOH global methane sink terms are each physically tenable, falling respectively towards the high- and low-end estimates of the Saunois et al. (2020) top-down budget assessment for 2008–2017 (501–574 Tg/y). The OH fields dominating methane removal in these two scenarios are likewise physically viable based on available independent constraints. If we attribute the optOH methane loss correction entirely to OH, we arrive at global-mean [OH] = $8.27 \times 10^5$ molecules/cm$^3$, placing the fixOH ($1.03 \times 10^6$ molecules/cm$^3$) and optOH solutions near the middle and lower end of the range indicated by prior assessments ($0.85$–$1.30 \times 10^6$ molecules/cm$^3$; (Krol and Lelieveld, 2003; Li et al., 2018; Montzka et al., 2011; Naik et al., 2013; Patra et al., 2021; Prinn et al., 2001; Prinn et al., 2005; Rigby et al., 2017; Zhao et al., 2019)). Our 2018–2019 analysis timeframe also spans an El Niño, which has been tied both to global OH decreases and to methane growth rate acceleration (Anderson et al., 2021; Turner et al., 2018a)—further complicating a differentiation between the fixOH and optOH solutions.

Additional ambiguity arises from the fact that the optOH methane sink adjustments could partly reflect uncertainty in the $CH_4$ + OH rate coefficient, which here follows Burkholder et al. (2020). While the rate at 298 K has been verified to within 1% across many lab experiments, uncertainties increase for higher and lower temperatures (up to 13% within 273–313 K (Burkholder et al., 2020)). Given all of the above considerations, we turn to independent measurements of methane, CO, and OH to assess the fixOH versus optOH solution fidelity.

## 3.2 Independent model assessments to discriminate between conflicting methane budgets

Ground-based methane column (XCH$_4$) observations from the TCCON network (GGG2014 (2014)) show comparable improvements over the prior for both the fixOH and optOH solutions (and for their individual member inversions), with 71% (from -12.9 ppb to 3.8 ppb) and 66% (to 4.3 ppb) mean bias reductions, respectively (Figure 3, Table S1). However, global in-situ measurements from ObsPack (near-real time version v2.0; (2021)) reveal a 93% (from -13.8 ppb to -0.9 ppb) absolute mean bias improvement for the fixOH framework compared to just 39% (to 8.4 ppb) for optOH (Figure 3, Table S1). Figure 3 further shows that the optOH solutions overcorrect the prior negative bias with respect to ObsPack, providing a first piece of evidence for a methane sink underestimate in this inversion.

Remote observations of OH and CO (the primary OH sink) from the ATom airborne campaign (Wofsy et al., 2018) also point to an OH underestimate in the optOH solution. Figure S9 compares the ATom observations for these species with predictions from supplemental GEOS-Chem simulations (configured as in Gonzalez et al. (2021)) constrained to the fixOH and optOH oxidant fields. With the exception of ATom 3, the mean model OH biases with respect to ATom observations are ~80% lower for fixOH than for optOH (mean differences are all significant based on a paired t-test at 95% confidence). These optOH results exhibit a consistent OH underestimate (averaging 0.020–0.044 ppt) that exceeds the 0.018 ppt measurement uncertainty. Biases in the simulated background CO levels are likewise lower (by 7–87%) in the fixOH simulations, with a clear CO overestimate for optOH (Figure S9). Again, the mean fixOH/optOH differences are all statistically significant at the 95% confidence level, with model-measurement discrepancies for optOH (7–12 ppb) exceeding the 3.6 ppb measurement uncertainty. While the ATom timeframe (2016–2018) is distinct from that of the TROPOMI inversions (2018–2019), the model is sampled at the time of measurement for both comparisons. We therefore expect the OH and CO biases highlighted above to likewise manifest for 2018–2019—an expectation that is supported by the ObsPack comparison.

When co-optimizing methane emissions and loss we thus find that the solutions can achieve a good fit to the TROPOMI data themselves but degrade model agreement with other observations of methane, OH and CO. We conclude that solving for global methane sources and sinks based solely on satellite observations of methane itself remains an under-constrained problem—even with the dense TROPOMI data coverage. In the same way, previous studies using methane data to optimize OH alongside emissions are likely subject to strong error correlations between the derived sources and sinks (Lu et al., 2021; Maasakkers et al., 2019; Qu et al., 2021; Zhang et al., 2018; Zhang et al., 2021). On the other hand, the comparisons here provide robust support for the fixOH inversion solutions based on their fidelity against independent atmospheric observations.

We thus focus the remainder of this paper on the fixOH results, and treat the corresponding multi-model mean as our base-case solution. The fixOH constituent inversions (Scale Factor - SF, Background Increment - BI, Observational Guess - OG, and Emission Enhancement - EE) each employ an alternative framework for spatial emission correction, as described earlier. Figure S10 shows that these individual members converge closely (within 0.5%) in terms of the total derived methane flux, with a high degree of spatial similarity in their derived emissions. Differences in adjustment magnitudes emerge, with the OG and EE frameworks generally yielding the largest and smallest emission corrections, respectively. Overall, the derived grid-level emissions agree to within 30% for 42% of the emitting model grid cells, with consistent adjustment direction over areas encompassing 65% of the total optimized emissions. In what follows we focus discussion on these areas of consistency, and use the multi-model spread to draw insights into some of the key disparities.

## 4. Global Methane Sources and Top Emitting Countries

We infer from the TROPOMI observations an optimized global methane flux of 587 (586–589) Tg/y for 2018–2019, including 375 (367–387) Tg/y from anthropogenic sources, 197 (185–207) Tg/y from natural sources, and 15 (14–15) Tg/y from biomass burning. Values listed reflect the fixOH multi-model mean and range, with emissions partitioned to individual sources according to the prior grid-level sectoral fractions. Our derived natural source falls at the low end of the 2017 Global Methane Project estimates (GMP; 194–489 Tg/y), whereas we infer a larger source from agriculture and waste than does GMP (253–262 versus 198–246 Tg/y) (Jackson et al., 2020). The TROPOMI-derived methane emissions from fossil fuel and industry of 118 (112–127) Tg/y lie between the GMP top-down (91–121 Tg/y) and bottom-up (121–164 Tg/y) estimates for that sector. Jackson et al. (2020) attributed the global methane emission increase between 2000–2006 and 2017 mainly to anthropogenic sources, with similar contributions from agriculture/waste and fossil fuel. Here, we find that emissions from agriculture and waste are even larger than estimated by GMP.

Table 2 lists the top ten national contributors to global anthropogenic methane emissions (see Table S2 for natural and total emissions). Together, we find that these contributors account for 58% of the global anthropogenic flux, and they similarly represent 60% of the global population. However, within this group there are large differences in per capita emissions, which are 72–286% higher than the global average in Brazil, the US, Russia, and Iran, but 17% lower in China and 45% lower in India. Table 2 also includes three of the five countries with the largest natural methane emissions, largely from wetlands: Brazil (39 Tg/y), the US (16 Tg/y), and Russia (12 Tg/y). Together with the Republic of the Congo (22 Tg/y) and Canada (15 Tg/y) these countries account for 53% of natural methane sources globally. This places Brazil (59 Tg/y) and the US (43 Tg/y) as the second and third methane emitters in terms of total flux, after China (61 Tg/y, 94% anthropogenic; Table S2).

Eight of the ten nations in Table 2 (China, India, US, Russia, Brazil, European Union, Pakistan, Indonesia) are likewise identified by Worden et al. (2022) as among the top ten anthropogenic emitters globally. Our inferred anthropogenic fluxes

for the US and China agree well (within ~10%) with the GOSAT-based results from Worden et al. (2022) and with the GOSAT+TROPOMI results from Qu et al. (2021). Anthropogenic emissions derived here are likewise within 10% of the Worden et al. (2022) results for India and the European Union, with both studies lower (20–50%) than Qu et al. (2021). Our results for Russia and Iran are 21–28% higher than the GOSAT-based estimates, mainly reflecting oil, gas, and coal emissions, and ~40% lower for Brazil, mainly due to livestock. Emissions for Pakistan and Indonesia agree to within 1% for the TROPOMI- and GOSAT-based results (Worden et al., 2022). However, we find here that anthropogenic emissions from Bangladesh (7 Tg/y versus a prior of 4 Tg/y) are 3× higher than the GOSAT estimate (2 Tg/y), while adjacent emissions from Myanmar (4 Tg/y) are half the GOSAT estimate. Worden et al. (2022) conclude that the GOSAT-derived emissions for Myanmar are anonymously high due to impacts from their prior assumptions; we attribute much of that flux to Bangladesh and show later that it mainly arises during the South Asian monsoon.

## 5. Wetland Sources are Underestimated in the Tropics

The TROPOMI-derived wetland fluxes (excluding rice) total 173 (155–182) Tg/y globally, representing 29 (26–31)% of the total methane source and 88 (84–91)% of the natural source. Figure S3, S11 and Figure 4e show that global wetland emissions are lowest during Oct–Feb (12–13 Tg/month) and highest in July (17 Tg/month) due to strong northern-hemisphere seasonality. We find through the inversions that global wetland fluxes are 24 Tg/y higher than the prior estimate, with the increase mainly originating in the tropics (82% within ±23.5° latitude). The tropics thus account for 70 (68–72)% of our optimized wetland emissions. Northern temperate wetlands contribute most of the remainder (46 Tg/y from 23.5°N–66.5°N) with a magnitude that is in-line with the prior bottom-up estimate (44 Tg/y). Our derived global wetland fluxes are ~20% higher than previous GOSAT-based estimates (145–148 Tg/y: Ma et al., 2021; Zhang et al., 2021), with similar latitudinal distribution to that found by Ma et al. (2021).

Over Amazonia (box 5 in Figure 2b), we obtain wetland fluxes of 51 (44–54) Tg/y, 29% of the global wetland total. These fluxes are underestimated in the WetCHARTs prior inventory by 9 (2–11) Tg/y; the true disparity is likely even larger since the inversions do not fully mitigate the prior regional XCH$_4$ bias (Figure 2c). Low observation density due to clouds (Figure S1) leads to some ambiguity in the spatial distribution of these derived fluxes: the SF and BI inversions allocate the upward corrections according to the prior spatial patterns over Amazonia, while the OG inversion identifies broader sources extending to northern Brazil (Figure 4 and S10). Upward corrections occur mainly during the wet season (Dec–Apr) and are temporally correlated with runoff (mean monthly R = 0.85 (ERA5, 2019)). This strong dependence on hydrology is likewise seen in the GOSAT-inferred flux increases over Amazonia that has been linked to increased flooding with strengthening Walker circulation (Barichivich et al., 2018; Zhang et al., 2021). However, eastern Brazil is also an agricultural frontier with forests transitioning to agricultural lands (Nepstad et al., 2019; Zhang et al., 2021), and our inversions point to a 9 (5–15) % underestimate of Amazonian livestock sources. Bottom-up calculations suggest a 33% increase in this source from 2010–

2018 (EDGAR v6, 2021); if such trends are in fact underestimated then our prior-based partitioning would imply an even larger livestock contribution to the derived regional emission corrections.

The inversions also point to a substantial (26 [5–36]%) upward correction for central African wetlands (box 12 in Figure 2b) that is concentrated during Dec–May (Figure S12). The optimized regional emissions are then 33 (28–36) Tg/y, 19% of the global wetland total. However, while these adjustments effectively correct the prior model bias for this latitude band, there is a clear XCH$_4$ over-correction for the Democratic Republic of the Congo (DRC; Figure 2c). Here, the modeled XCH$_4$
enhancement over the background (Figure S13) is degraded from a prior model-observation mismatch of +3 ppb to +11 ppb. The low data density over central Africa and Amazonia (Figure S1) thus appears to cause some tropical flux mis-attribution—with the DRC overcorrection offset by under-corrections over Amazonia (as discussed above), Nigeria, and the nearby Sudd wetlands (Figure 2c). The latter region is examined further below.

Figure 2a shows that the South Sudd wetlands (box 13 in Figure 2b) are a major methane hotspot that is underestimated in the prior model by a column average of 41 (21–65) ppb. Despite this, we derive a regional upward emission correction of just 13% (optimized flux: 1.3 [1.2–1.4] Tg/y). This yields a residual underestimate (Figure 2c) reflecting the aliasing discussed above and showing that the optimized South Sudd fluxes are still too low. Anomalous hydrology may contribute to these elevated Sudd fluxes: wetland extent for this area was ~10% higher in 2019 than the 2010–2019 mean (Jensen and
Mcdonald, 2019), and ERA5 reanalysis (2019) points to elevated precipitation over central Africa during Sep–Nov (22% above the 2010–2019 mean). This interpretation is consistent with a previous study by Lunt et al. (2021) linking anomalous East Africa rainfall during the 2018 long rains (Mar–May) and 2019 short rains (Oct–Dec) with 10–40% methane emission increases over South Sudd. Over longer timescales, GOSAT analysis has pointed to a 3 Tg/y emission increase over the broader Sudd region caused by increased inflow from the White Nile and Sobat rivers (Lunt et al., 2019; Maasakkers et al.,
2019; Parker et al., 2020). Previous SCIAMACHY and TROPOMI-based analyses have likewise identified concentration hotspots over this area (Hu et al., 2018; Frankenberg et al., 2011).

North American wetlands in Alberta, Saskatchewan, and the US Upper Midwest are revised downward (by -1.6 [-0.7 – -2.6] Tg/y for box 1 in Figure 2b), with most of the adjustment occurring during early summer (-18% for Jun-Jul vs. -1% for Aug–
Sep; Figure 4). We thus obtain optimized regional wetland emissions of 16 [15–17] Tg/y, 9% of the global flux from this source. Wetland emissions over this area are highly sensitive to soil temperature and surface water extent (R > 0.7 and R > 0.4, respectively; Figure 5). The stronger Jun–Jul adjustment derived here suggests that the post-thaw onset of northern wetland fluxes occurs too early in WetCHARTs, which uses surface skin rather than soil temperatures to predict emissions. Similar downward emission corrections have been inferred from GOSAT (Zhang et al., 2021), aircraft, and long-term eddy
covariance measurements (Yu et al., 2021c).

Across the wetland regions examined above, Figure S14 shows that our optimized emissions fall towards the middle of the land-surface model estimates from the Global Carbon Project (GCP; details on these bottom-up models and their differences are provided by Saunois et al. (2020)). Among the GCP bottom-up estimates, we see in Figure S14 that simple model parameterizations can obtain comparable agreement with the TROPOMI-optimized emissions as more complete process-based models. For example, LPJ-WSL, a parsimonious model that predicts net emissions based on soil characteristics without explicitly representing oxidation, transport, or wetland plant types, achieves similar fidelity (in terms of bias and RMSE versus the optimized values) as JSBACH (Kleinen et al., 2020), which features more sophisticated treatment of soil carbon, roots, and plant-mediated processes. Many simple wetland models rely on reanalysis-based wetland extent estimates such as the Global Lakes and Wetlands Database (GLWD) (Lehner and Döll, 2004); within the WetCHARTs ensemble we find here that GLWD-based flux predictions overestimate emissions in northern North America while underestimating those in central Africa (Figure S15).

Figure 5c explores the environmental sensitivities of the TROPOMI-derived wetland sources to gauge how future rainfall or temperature changes may alter emission magnitudes, and to motivate further analyses. We see that the optimized tropical and subtropical wetland methane emissions exhibit only modest sensitivity to soil temperature (0–7 cm (ERA5, 2019)), but have strong (R > 0.7) correlation with surface water extent (SWAMPS (Jensen and Mcdonald, 2019)) for key areas of India, Bangladesh, Brazil, Bolivia, Mexico, and Africa. Conversely, northern temperate wetland emissions in the US Upper Midwest and southeastern China respond more directly to temperature changes, while those in Canada and Russia show strong sensitivity to both hydrology and temperature. Projected precipitation increases for mid-to-high latitudes and decreases for the sub-tropics (IPCC, 2021) may thus increase the importance of temperate and boreal wetland fluxes in coming years.

## 6. Anthropogenic sources and emission hotspots

We derive global anthropogenic methane sources of 375 Tg/y, with 132 (127–136) Tg/y from livestock, 98 (93–104) Tg/y from fossil fuel, 83 (79–87) Tg/y from waste, 42 (40–45) Tg/y from rice, and 20 (19–23) from other sources. This represents a 19 Tg/y increase over the prior flux that on a global basis is mainly driven by upward corrections for livestock (+11%) and rice (+15%). The 2019 global anthropogenic methane emissions obtained here are modestly (12%) higher than GOSAT-based results for 2010–2018 (336 Tg/y; Zhang et al., 2021), with both results pointing to higher-than-predicted biotic emissions (consistent with isotopic constraints; Nisbet et al., 2016). Since our prior anthropogenic emissions are based on inventories for 2010–2016, the derived flux corrections could reflect inventory errors, temporal changes between 2010 and 2019, or some combination of the two. Below, we employ the spatially-downscaled TROPOMI-derived emissions to elucidate key anthropogenic sources and to identify missing and underreported flux hotspots.

### 6.1 The Middle East and North Africa: Missing and underestimated emission hotspots from fossil fuel activities

The largest fossil fuel emission corrections occur over the Middle East, where total fluxes increase throughout the year by 48 (34–60)% over the prior (+12 [9–16] Tg/y, box 9 in Figure 2b). These upward corrections successfully reduce the mean regional model bias from -29 to 0 ppb and are attributed to a combination of fossil fuel (41%), livestock (23%) and waste (20%). The Middle East possesses approximately ~50% of global oil reserves and ~40% of global natural gas reserves, with increasing production over the past three decades (BP, 2021; Schneising et al., 2020; UNFCCC, 2021). Bottom-up information (EDGAR v6, 2021) accordingly points to a significant increase in methane hotspots for this area over the recent decade (Figure 5a) and to an overall 26% regional emission increase from 2010 to 2018. The 1.8× larger adjustment revealed here by the TROPOMI observations points to both temporal increases and inventory underestimates for this area.

Middle East methane emissions are highly localized, with just 5% of model grid cells (at 0.1°×0.1°) accounting for 70% of the prior regional emissions and 60% of the derived adjustments. Our multi-model inversion results point to consistent prior underestimates for high emitters in Azerbaijan, Turkmenistan and Iran, supporting previous satellite-based analyses (Buchwitz et al., 2017; Lauvaux et al., 2022; Schneising et al., 2020; Varon et al., 2019). However, TROPOMI also reveals a large number of hotspots (over Oman, Yemen, Saudi Arabia, Iraq, Turkmenistan, and Iran; Figures 2a, 5b, and S16) that are entirely missing from the prior inventory. Many of these missing emission hotspots are consistent with facilities on the ground. For example, missing sources in Oman identified through the OG inversion (Figure S10 and S16) match the location of the Khazzan gas field—one of the Middle East's largest natural gas fields producing ~1–1.5 billion cubic feet/day of natural gas and ~35,000 barrels/day of light oil (NS Energy, 2022). Other detected hotspots correspond to the Masila Basin in Yemen (EIA, 2022), oil fields in Saudi Arabia (e.g, the Ghawar Field, (Maps Saudi Arabia, 2022)) , and super-emitters in Iraq (Lauvaux et al., 2022).

Over northern Africa, TROPOMI identifies large $XCH_4$ enhancements extending from the Libyan coast to Algeria (box 8 in Figure 2b). These sources are not well-represented in the prior inventory but correspond to oil fields in Libya and to part of the Greenstream pipeline system. The OG inversion is partially able to identify these sources, but the attribution is limited to a single 2°×2.5° grid cell (Figure S10).

### 6.2 Western Russia and North China Plain: An overestimate of fossil fuel emissions

We find from the TROPOMI data that emissions are overestimated in northern Asia, mainly reflecting fossil fuel sources. Specifically, fossil fuel methane emissions are overestimated by 27 (11–40)% over western Russia (box 6 in Figure 2b) and by 17 (1–29)% over the North China Plain (box 7), with these downward corrections reducing the regional model bias to <6 ppb. The UNFCCC-2016 emissions used as prior account for accidental and intentional methane releases not considered in previous inventories (Scarpelli et al., 2020), leading to a >2-fold increase over western Russia compared to EDGAR v5. Our

results indicate that these emission pathways may be overestimated in UNFCCC-2016, and that the actual fossil fuel methane source from Russia for 2018-2019 lay between the UNFCCC-2016 and EDGAR v5 values. Indeed, subsequent revisions (year-2019; Scarpelli et al., 2022) to the UNFCCC-2016 inventory used here have strongly reduced fossil fuel emission estimates for Russia (e.g., 21 Tg/y from oil in year-2016 vs. 2 Tg/y in year-2019) due to updated emission factor assumptions.

Methane emissions in the North China Plain are primarily from coal mines; prior work suggests that these facilities have lower emission factors than in South China and that their emissions have been declining since 2012 (Sheng et al., 2019). Satellite-based and in-situ measurements have pointed to EDGAR v4.3.2 and GFEI emission overestimates for this area (Alexe et al., 2015; Lu et al., 2021; Maasakkers et al., 2019; Monteil et al., 2013; Qu et al., 2021; Turner et al., 2015; Zhang et al., 2021). Here, we use a detailed new inventory for China coal emissions (Sheng et al., 2019) that has >50% lower fluxes than EDGAR v4.3.2 over the North China plain—but find that these are still overestimated. By contrast, upward adjustments are derived over other regions in China such as Xinjiang, where TROPOMI $XCH_4$ enhancements are attributed by the OG and EE inversions to underestimated fossil fuel sources (Figure S10). Recent work by Lorente et al. (2022) points to some erroneous emission hotspots for this area associated with surface reflectance; we therefore restrict our interpretation here to the regional scale. Positive corrections are also derived for fossil fuel sources in South China, but their emissions are difficult to isolate from those of nearby rice fields.

### 6.3 South and Southeast Asia: Major methane emissions during summer monsoon

Methane emissions from South and Southeast Asia are dominated by agriculture (livestock, rice) and waste. The largest emission correction for this area occurs over India, where we find a 23 (10–30)% underestimate (mainly due to those two sources) and derive total national emissions of 61 [55–64] Tg/y. India contains over 35% and 20% of the world's cattle and water buffalo, respectively, with both populations increasing over the past decade (Sonavale et al., 2020). Such changes are reflected in both the EDGAR bottom-up inventory (+0.4 Tg/y annual livestock+waste emission increase for 2010–2018 (EDGAR v6, 2021)) and in analyses based on GOSAT observations (+0.2–0.7 Tg/y inferred annual increase for Indian livestock (Maasakkers et al., 2019; Miller et al., 2019; Zhang et al., 2021)). Our inversion for 2018-2019 uses an EDGAR v5 prior estimate for year-2015, but the inferred +23% (11 Tg/y) correction is too large to be explained solely by intervening trends and thus indicates an emission underestimate for this source.

Over 90% of the world's rice production occurs in India and Southeast Asia, and we find that the associated methane emissions are underestimated by 39 (7–53)% and 17 (7–28)% respectively (boxes 10 and 11 in Figure 2b). Bottom-up statistics from FAOSTAT (2021) indicate an 8% increase in crop production over these areas for 2010–2019 due to a 10% yield increase combined with a 2% cropland area decrease. Expected emission trends between the timeframe of the EDGAR v5 emissions (2015) and our inversions (2018-2019) can therefore not fully explain the TROPOMI observations, indicating

that the prior emissions for this source are too low. The emission corrections mainly occur during the Jul–Oct rice growing season that coincides with the summer monsoon. The TROPOMI data reveal several other connections between the East Asian summer monsoon and the regional methane budget, which we explore next based on our monthly downscaled top-down emissions.

Approximately 80% of India's annual rain falls during the summer monsoon (IPCC, 2021), and across this Jul-Oct season we find that methane emissions from the India and Southeast Asia boxes in Figure 2b are underestimated by 37 (15–45)%. The resulting seasonal flux increase then accounts for over 68% of the total annual emission correction. While clouds reduce the TROPOMI sampling coverage during this time, we still obtain >370,000 and >9,000 observations per monsoon season over India and Southeast Asia, respectively, after applying the data quality filters described in Section 2.1. The above emission corrections strongly reduce the prior model biases (from -28 ppb to 3 ppb over India and from -41 ppb to -7 ppb over Southeast Asia), with the individual inversions pointing to consistent spatial adjustments. Prior work using GOSAT and in-situ measurements has also identified a peak in Indian emissions during Jul–Oct that was not well-captured by bottom-up predictions (Palmer et al., 2021).

Figure 6 shows that the TROPOMI-derived South Asian emission corrections are spatially and temporally coherent with the summer monsoon onset and withdrawal. Strong upward adjustments are first derived over Bangladesh and East India following monsoon arrival over these areas (early June 2018; late June 2019). As the monsoon advances, the emission corrections then extend more broadly over northern India during Jul–Sep. In concert with the monsoon, the upward corrections subsequently withdraw south and east to Bangladesh in October. Only minor emission adjustments are derived for this region outside of the summer monsoon.

The above patterns likely reflect hydrologic influences on methane emissions from biotic sources such as wetlands, rice fields, manure, landfills, and sewers. The derived emission corrections exhibit a strong temporal correlation with runoff (R = 0.82 in eastern India; R = 0.62 in western India), supporting the underlying role of hydrology. Baker et al. (2012) similarly concluded on the basis of aircraft measurements that 64% of Indian methane emissions during this season reflect monsoon-driven biogenic sources. Increases in Indian summer monsoon precipitation that are projected over the 21th century (Katzenberger et al., 2021) thus raise the strong possibility of enhanced regional methane emissions in the years to come.

**6.4 North and South America: Increases for fossil fuel sources**

We turn next to key oil and gas production fields in North and South America. This includes the US Permian, Barnett and Eagle Ford region (box 2 in Figure 5b), previously shown to be the largest US oil and gas-related methane source (Zhang et al., 2020). Here we infer methane fossil fuel emissions for 2018–2019 that are within 2% of the prior GEPA estimate. The GEPA estimates are for year-2016, but based on EDGAR v6 there was no significant regional trend between 2016 and the

2018-2019 inversion period. Over northern Venezuela (box 4 in Figure 5b), we derive a 28 (2–41)% upward correction for fossil fuel exploration activities that is consistent across the year, in agreement with the +32% bottom-up increase for that source that is estimated to have occurred between the prior inventory and inversion timeframes (2016–2018; (EDGAR v6, 2021)).


Extensive TROPOMI XCH$_4$ enhancements are seen over southern Mexico (Figure 2b, box 3), and our inversions reveal a 15 (9–25)% emission underestimate for this area (optimized flux 10 [9–10] Tg/y). Trend information from EDGAR v6 (2021) suggests a regional 25% emission increase for 2016–2020, which could in theory fully explain the derived upward adjustments. However, the observed hotspot locations are largely missing in the prior inventory, and as a result the SF

inversion falsely attributes the corrections to upwind waste/landfill sources in Sinaloa. The OG inversion solution aligns more closely with the actual locations of these sources based on the oversampled TROPOMI data (Figure S10), and is supported by previous regional-scale TROPOMI inversions (Shen et al., 2021) that point to a >2× emission underestimate of onshore/offshore oil and gas production in the UNFCCC-2016 inventory used here. Aircraft measurements in 2018 also revealed substantial (29,000 kg/h) methane emissions in the same general region (Zavala-Araiza et al., 2021).

**7. Conclusions**

A suite of two-year 4D-Var inversions using satellite-based data from TROPOMI places new constraints on global methane sources. We obtain in this way optimized global emissions of 587 (586–589) Tg/y for 2018–2019. Compared to the most recent GCP estimates (Jackson et al., 2020), our 2018–2019 results point to a larger role for anthropogenic sources, mainly tied to agriculture and waste. We further develop a new framework to map the derived monthly emissions to 0.1° × 0.1°

resolution, enabling the identification of key missing and underestimated sources as highlighted below.

We derive a +24 Tg/y increase in wetland emissions over the prior estimate of 149 Tg/y that mainly (82%) occurs over the tropics and appears to be related to positive hydrologic anomalies in Amazonia and the Sudd. Meanwhile, fossil fuel emissions in the Middle East are underestimated by 47 (23–57)% and reached 15.7 (13.2–16.8) Tg/y during our analysis

period. Our inversions further uncover missing emission hotspots over Turkmenistan, Iran, Oman, Yemen, Iraq, Libya, Algeria, and Mexico. We estimate long-standing fossil fuel sources in Venezuela at 4.8 (3.8–5.3) Tg/y, 28 (2–41)% higher than the prior estimate (which is for year-2016).

Inversions point to underestimated agricultural sources in India, the Amazon Basin, central Africa, the US California Central

Valley, and Asia. However, more than 45% (8.5 [3–11] Tg) of the global anthropogenic source adjustment derived here occurs over India and southeast Asia during the summer monsoon (Jul–Oct). We postulate that this reflects the influence of

monsoon rainfall on methane emissions from rice, manure, waste, and landfills. Given the projected increase in monsoon precipitation over the coming century (IPCC, 2021), better understanding of these effects is crucially needed.

Finally, our analyses show that even the dense TROPOMI data coverage does not fully resolve variability in methane sources from that in its sinks. We address the issue in this work by employing validated OH fields from a chemical transport model, but future methane inversions can benefit from incorporating additional datasets (e.g., CO, methyl chloroform, formaldehyde) as constraints on the methane sink (McNorton et al., 2016; Rigby et al., 2017; Turner et al., 2017; Wolfe et al., 2019). Quantitative evaluation of the influence of the 2018–2019 El Niño and the 2021 Hunga Tonga–Hunga Haʻapai

eruption on OH variability will also help to advance the accuracy of contemporary methane budget estimates.

**Code and data availability**

TROPOMI data is publicly available online at http://www.tropomi.eu/data-products/level-2-products. TCCON data was obtained from the TCCON Data Archive hosted by CaltechDATA at https://tccondata.org. The GEOS-Chem adjoint code is available at http://adjoint.colorado.edu:8080/gcadj_std.git. The modified code used here is archived at

https://doi.org/10.13020/g5xc-nj81 (Yu et al., 2021b).

**Competing interests**

The authors declare that they have no conflict of interest.

**Author contributions**

XY performed the 4D-Var inversions with help from DBM and DKH. XY, DBM, and DKH designed this study. All authors

contributed to the analyses. XY and DBM wrote the manuscript, with comments from all authors.

**Acknowledgements**

We thank Ilse Aben for assistance with the TROPOMI data; Lee Murray, Benjamin Poulter, Marielle Saunois, and Tia Scarpelli for helpful discussions; and William Brune, David Miller, and Alexander Thames for the ATom OH measurements. We also thank the TCCON, ObsPack, and ATom teams for providing observations used here. This work was

supported by NASA's Interdisciplinary Research in Earth Science program (IDS Grant #NNX17AK18G), by the US EPA (Assistance Agreement #R835873; Center for Air, Climate, and Energy Solutions (CACES)), and by the Minnesota Supercomputing Institute. XY acknowledges support from a NASA Earth and Space Science Fellowship (Grant #80NSSC18K1393). Part of this work was carried out at the Jet Propulsion Laboratory, California Institute of Technology,

under a contract with the National Aeronautics and Space Administration (NASA). The research has not been formally reviewed by the funders. The views expressed in this document are solely those of authors and do not necessarily reflect those of the funders. Funders do not endorse any products or commercial services mentioned in this publication.

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

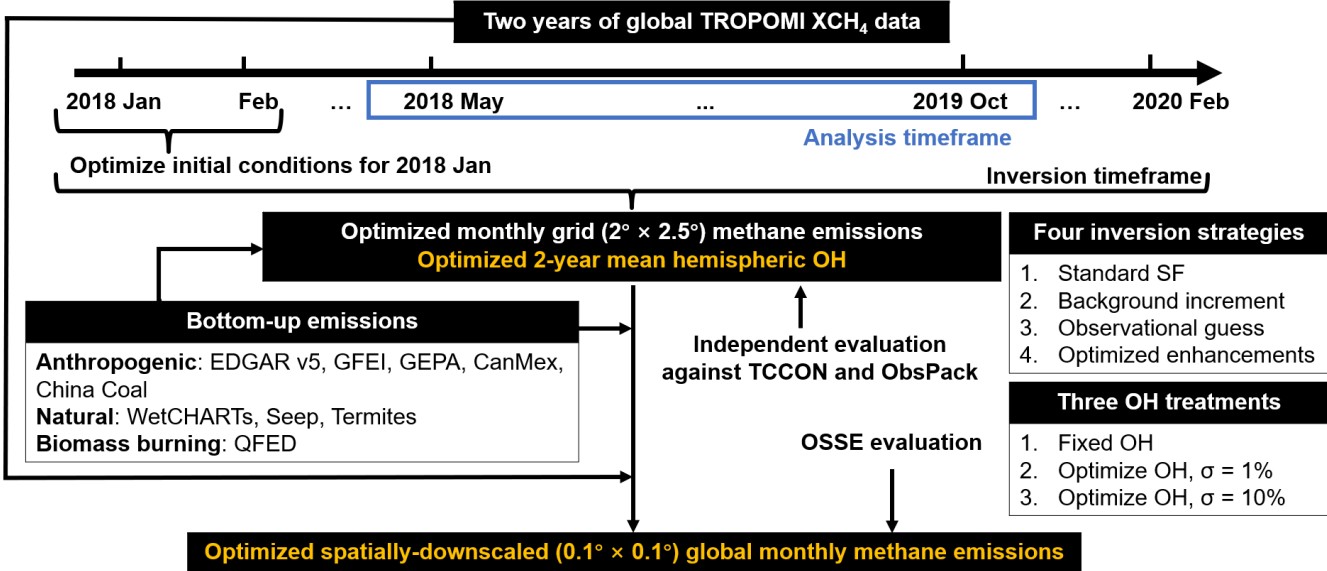

**Figure 1. Flow chart showing the TROPOMI methane inversion methodology.**

1060

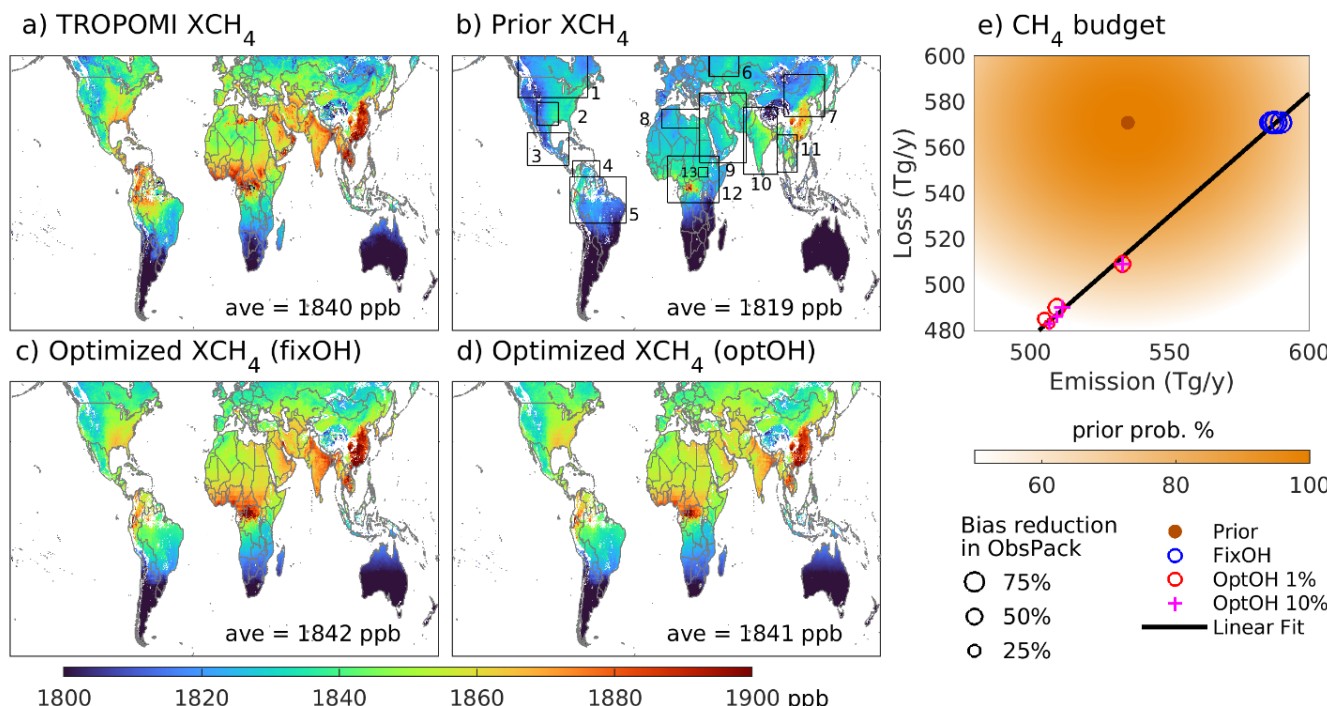

**Figure 2. Observed and simulated methane distributions for 03/2018–02/2020. a)** TROPOMI methane column (XCH₄) observations oversampled to 0.1° × 0.1°. **b)** Methane column distribution predicted by the prior model sampled according to the TROPOMI observation operator with numbered regions described in-text. **c)** Same as panel b but for the fixOH optimized ensemble mean. **d)** Same as panel b but for the optOH optimized ensemble mean. **e)** Prior and optimized methane budgets. Prior probabilities (orange) equate one standard deviation to 25% of the bottom-up range from Saunois et al. (2020), following Hozo et al. (2005). Black line shows a linear fit to the solution, emission = 0.93 × loss + 55.26 (units: Tg/y). Symbol sizes indicate the mean bias reduction against ObsPack independent measurements. Annual values are for 11/2018–04/2019 plus the average of 05–10/2018 and 05–10/2019.

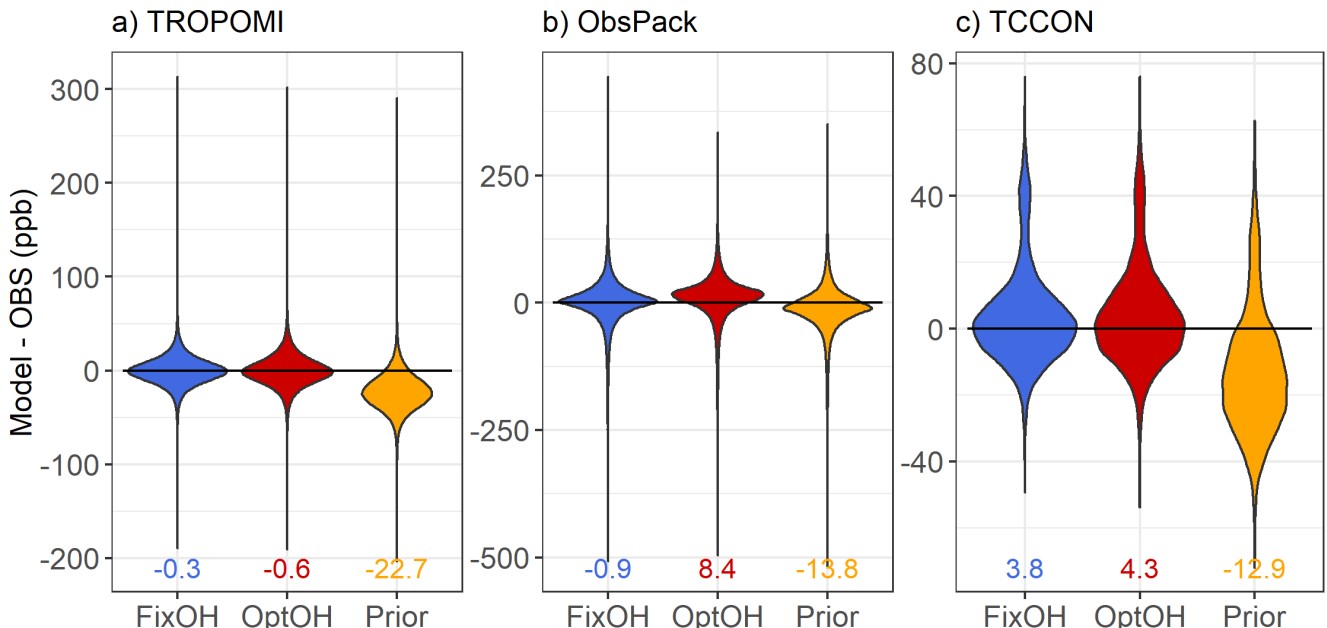

**Figure 3. Evaluation of the prior and optimized methane simulations (05/2018–10/2019) against a) TROPOMI XCH₄ observations, b) ObsPack in-situ methane measurements, and c) TCCON ground-based methane column observations. Numbers inset indicate the mean model-observation bias (unit: ppb).**

1075

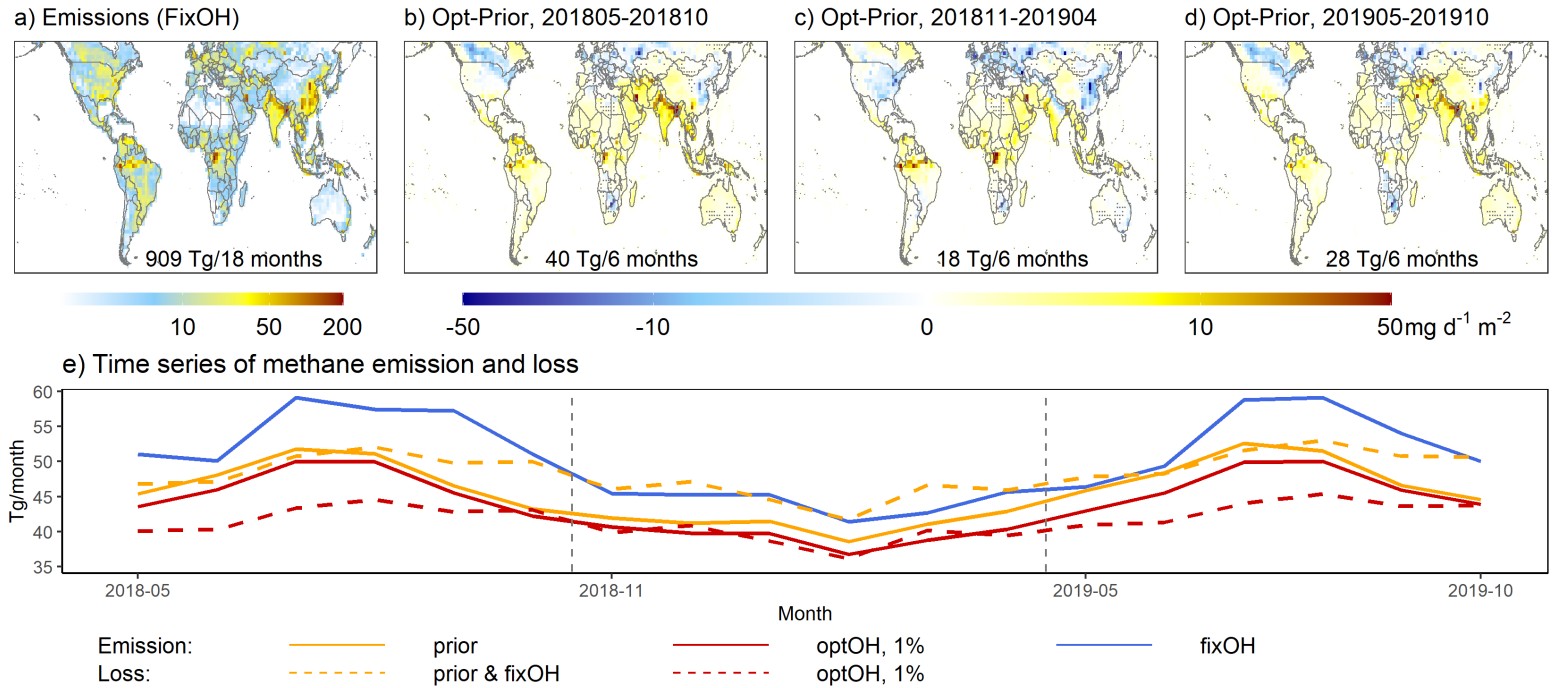

**Figure 4. a) Optimized global methane emissions based on the fixOH inversion ensemble. b)–d) Seasonal fixOH emission corrections. Dots indicate missing sources where the EE and BI inversions point to positive corrections that the SF inversion misses. e) Time series of prior and optimized methane sources and sinks.**

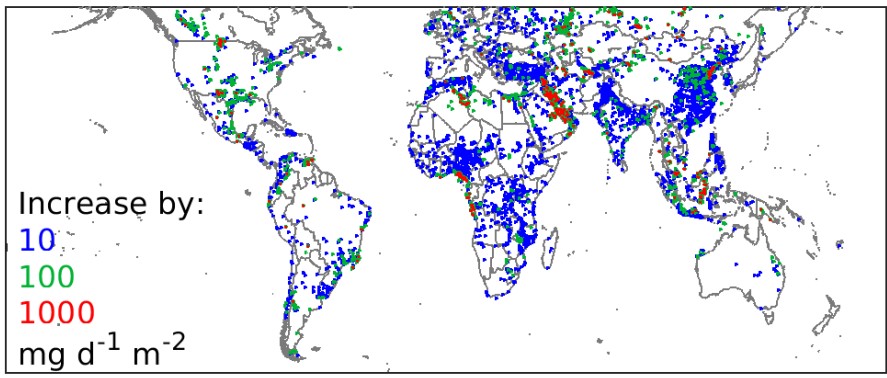

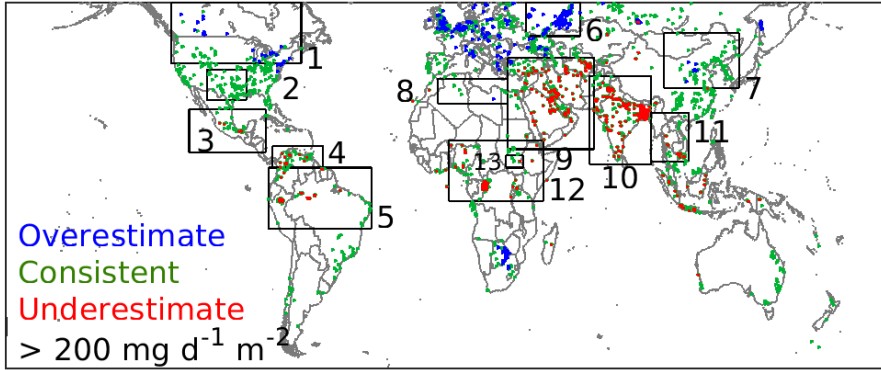

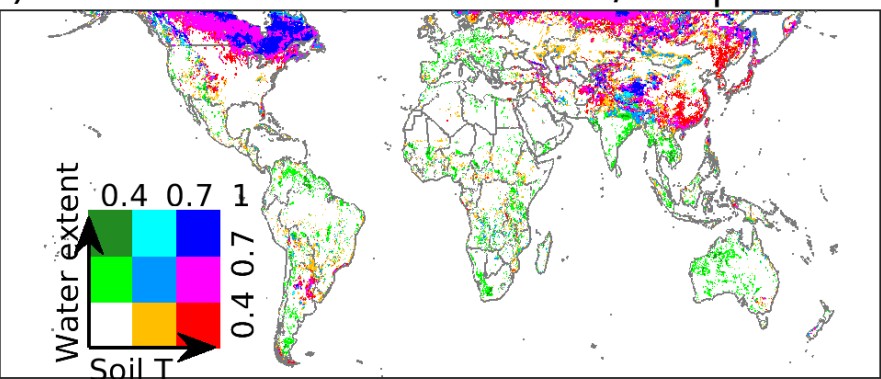

1080

**Figure 5. a) Methane emission increases during 2010–2018 estimated from bottom-up information (EDGAR v6, 2022). b) Methane source hotspots (>200 kg d[-1] km[-2]) and their TROPOMI-derived emission corrections (blue: >30% overestimate; green: accurate to ±30%; red: >30% underestimate). c) Correlations between the optimized wetland emissions, soil temperature (0-7 cm; ERA5 2019), and surface water extent (Jensen et al., 2019).**

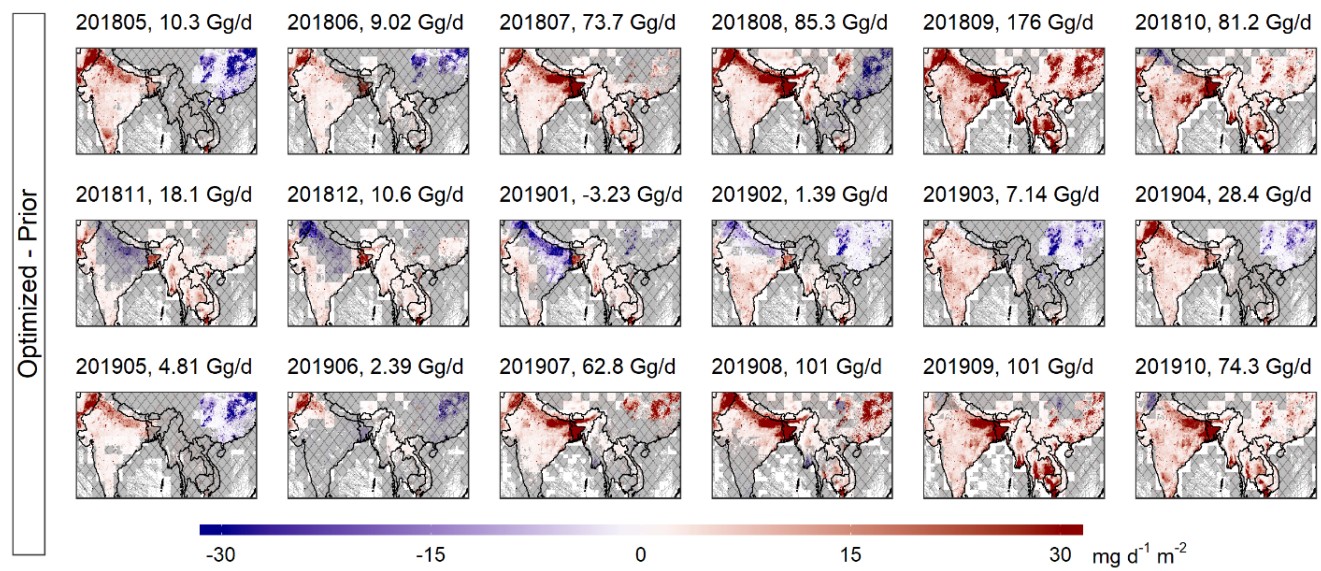

**Figure 6. TROPOMI-derived emission corrections over South and East Asia based on the fixOH ensemble mean. Shading indicates areas where corrections are not distinguishable from zero (i.e., inversion ensemble includes both positive and negative adjustments).**

**Table 1. Downscaling performance evaluation via OSSE[1]**

| | Downscaled solution | Fine-grid prior | Fine-grid adjoint | Coarse-grid prior | Coarse-grid adjoint |
|---|---|---|---|---|---|
| **Without transport error** | | | | | |
| Domain-wide bias reduction | 68% | | 78% | | 68% |
| RMSE[2] | 15790 | 13830 | 11489 | 13236 | 12976 |
| R | 0.67 | 0.57 | 0.74 | 0.45 | 0.50 |
| Slope of least squares fit | 0.97 | 0.61 | 0.84 | 0.28 | 0.42 |
| **With transport error** | | | | | |
| Domain-wide bias reduction | 98% | | 16% | | 98% |
| RMSE[2] | 15461 | 13830 | 19292 | 13236 | 12917 |
| R | 0.70 | 0.57 | 0.60 | 0.45 | 0.51 |
| Slope of least squares fit | 1.04 | 0.61 | 1.01 | 0.28 | 0.45 |

[1.]Observing system simulation experiments performed for one month over North America

[2.]Root-mean-square-error, units: kg d$^{-1}$ box$^{-1}$

**Table 2. Top 10 contributors to global anthropogenic methane emissions[1,2]**

| | Total anthropogenic emissions (Tg/y) | Change from prior (%) | Per capita anthropogenic emissions (kg/y/person) | Sector emissions (Tg/y) | | | |
|---|---|---|---|---|---|---|---|
| | | | | Wetland | Agriculture & waste | Fossil fuel | Other |
| China | 57 (52–61) | -4 | 40 | 2 (2–2) | 34 (32–36) | 18 (16–21) | 6 (6–6) |
| India | 36 (34–38) | 16 | 26 | 2 (2–3) | 32 (30–33) | 2 (2–2) | 3 (3–3) |
| US | 27 (26–28) | -5 | 82 | 15 (14–15) | 16 (15–17) | 11 (10–11) | 2 (1–2) |
| Russia | 26 (22–29) | -21 | 184 | 10 (8–12) | 5 (5–6) | 21 (17–24) | 2 (2–4) |
| Brazil | 20 (20–21) | 6 | 102 | 35 (31–37) | 20 (19–20) | < 0.2 | 4 (4–4) |
| European Union | 17 (15–17) | -9 | 38 | 2 (1–2) | 14 (12–14) | 2 (2–2) | 2 (2–2) |
| Pakistan | 10 (9–10) | 28 | 44 | < 0.1 | 9 (8–9) | 1 (1–1) | 1 (1–1) |
| Indonesia | 10 (9–10) | 15 | 37 | 9 (8–10) | 8 (8–8) | 1 (1–1) | 2 (2–2) |
| Iran | 9 (7–9) | 70 | 103 | < 0.2 | 2 (2–3) | 6 (4–7) | 1 (0–1) |
| Bangladesh | 7 (5–8) | 60 | 43 | 1 (1–1) | 7 (4–8) | < 0.1 | <0.4 |
| Others | 157 (151–168) | 13 | 50 | 97 (87–103) | 111 (108–118) | 37 (36–38) | 37 (30–46) |

[1]Based on the fixOH inversion ensemble mean and range.
[2]Values in parentheses indicate the range in emission estimates across the suite of inversions.