# Peer review of "A high-resolution satellite-based map of global methane emissions reveals missing wetland, fossil fuel and monsoon sources"

_EGUsphere, 2022_

## Author Comment (AC1)

**Reply to Review 1.**

We thank the reviewer for the constructive comments. Reviewer comments are provided below in black with our responses in blue.

The paper is mostly well written, except for the section comparing posterior OH to ATOM results. The main concern I have is that the method and results in this paper are not fundamentally different from those in the Qu et al. ACP 2021 paper which also uses TROPOMI data for quantifying emissions for essentially the same time frame. The main difference between this paper and the Qu et al. paper is the version of the data, which is ostensibly more accurate than the data used in Qu et al. but then there is no discussion on how this improved data set changes, or potentially improves the results over and above the Qu et al. results.

For acceptance, the paper needs to better describe the difference from those in Qu et al, and how these results are an improvement ; I think there are sufficient results in here for this purpose (e.g. monthly estimates allow for attributing some components of the methane budget). In addition, you could compare with the Qu et al. 2022 paper (methane surge) which uses GOSAT data for 2019; in principal the improved TROPOMI data sets should result in better comparisons with the GOSAT based results for this time period.

We have revised the manuscript as follows to better highlight the differences between our study and that of Qu et al.

Section 2: " Recent inverse analyses by Qu et al. (2021) likewise examined the global methane budget using TROPOMI (and GOSAT) observations. Our study advances on that work in several ways. First, we optimize monthly rather than annual fluxes to identify seasonal patterns of variability. Second, in place of a traditional analytical optimization we combine 4D-Var with new inverse formalisms for better identification of missing sources. Third, we develop a new downscaling approach to constrain emissions at high resolution, and use this framework to elucidate flux mechanisms and missing sources. Finally, our analysis leverages an updated TROPOMI methane product (Lorente et al., 2021) that corrects an albedo-dependent bias present in the version used by Qu et al. (2021)."

As the reviewer suggests, we have also revised the draft to provide more details on the updated data version employed here:

Section 2: "Relative to the albedo-corrected product, the prior TROPOMI version exhibits high biases over North Africa, the Middle East, and the western US, and low biases over Amazonia, the eastern US, central Africa, and eastern China (Lorente et al., 2021)."

Another issue is a lack of discussion on uncertainties; I see them reported in final estimates but its not obvious how they are computed, are these buried in the text somewhere ?( Im pretty sure I read through the entire text, 2.5 times + browsing). A more extensive discussion on uncertainties should be in Section 2.

This is now explained more prominently in the text:

Section 2: "In the following, we interpret the multi-inversion mean as our base-case solution and the range as the corresponding uncertainty estimate."

**We have also added a new analysis to further characterize the downscaled solution accuracy using the OSSE described in-text:**

**Section 2: "Figure S8 shows that the downscaled OSSE solution reduces the prior bias by 17%–56% for sources exceeding 1000 kg CH$_4$/box/day (accounting for 99% of the domain-wide emissions) when not subject to transport error. In the presence of transport error, the downscaling method has limited success for the very largest sources (>2×10$^5$ kg/box/day), but nevertheless exhibits strong bias reduction (21%–50%) for sources between 1×10$^3$–2×10$^5$ kg/box/day (96% of domain-wide emissions)."**

[Figure]

**Figure S8. Downscaling bias reduction as a function of emission magnitude, based on 1-month Observing System Simulation Experiments (OSSE) over North America (see main text for details).**

Note that Im not convinced that the downscaling approach described here is sufficient by itself to merit publication as it is not (obviously) an improvement over the optimal estimation based approach described in the un-cited Cusworth et al. 2021 paper (see subsequent comments).

**We strongly disagree with this perspective. The downscaling method developed and applied here is a novel contribution and uses an entirely different strategy from that of Cusworth et al. (2021). Cusworth et al. (2021) combine the posterior and prior uncertainties to obtain sector-based emission estimates. Our method takes a different approach, combining the prior error estimates directly with the (oversampled) TROPOMI observations themselves to project emissions from coarse to fine resolution. A unique advantage of our approach lies in this use of the sub-model-grid satellite information (which is lost in a standard lower-resolution inversion) to inform the downscaling. Further details are provided in our replies to the more specific comments below.**

**We now compare our method with that of Cusworth et al. (2021) in text as follows.**

**Section 2: "Compared to existing emission downscaling methods that rely on prior and posterior error covariance estimates (Cusworth et al., 2021), or are based solely on satellite data (Liu et al., 2021), our approach is unique in combining the prior emission information (and its uncertainty) with the oversampled TROPOMI observations themselves. Variable weighting between these terms permits greater influence from the observations when the prior emissions are more uncertain. The method**

**thus assumes robust prior error estimates, a caveat that also applies to Cusworth (2021) and similar methods."**

Specific Comments:

Abstract, "… indicate rapid increases in Middle East"; the way this sentence is currently written implies you base this statement on satellite observations.

**We have clarified the Abstract as follows:**

**"Emissions from fossil fuel activities are strongly underestimated over the Middle East (+5 Tg/y), where bottom-up inventories suggest rapid increases over the past decade, and over Venezuela."**

Abstract: state that you are estimating monthly values

**We have revised the Abstract as suggested:**

**"We apply a new downscaling method to map the derived monthly emissions to 0.1°×0.1° resolution, using the results to uncover key gaps in the prior methane budget."**

Abstract: You stated you used observations of CO and OH, but I don't see any description of these data in Section 2. Also see comments on comparisons to CO and OH to ATOM below

**We have revised the abstract to clarify that the CO and OH data are used for evaluation:**

**"Employing remote carbon monoxide (CO) and hydroxyl radical (OH) observations with independent methane measurements for evaluation, we infer from TROPOMI a global methane …."**

**We have also added a description of the evaluation datasets to Section 2, as suggested:**

**"We use a large suite of independent measurements to evaluate the inversions. These include methane columns from TCCON (2014), a global network of Fourier transform spectrometers, and methane mole fractions from ObsPack (near-real time version v2.0; (2021)), a global compilation of ground-based and airborne measurements. We further use CO and OH measurements from the Atmospheric Tomography (ATom) airborne campaign (Wofsy et al., 2018) to test inversion success at separately optimizing methane sources and sinks. ATom featured pole-to-pole profiling (0.2 to 12 km) during four seasons over four years. The flight design is thus well-suited to determine whether the optimized OH fields improve or degrade global model simulations of OH itself and of CO (whose dominant sink is reaction with OH). Measurements of CO during ATom were performed using the NOAA Picarro instrument with an estimated uncertainty of 3.6 ppb (Chen et al., 2013). OH measurements during ATom employed the Airborne Tropospheric Hydrogen Oxides Sensor (ATHOS), with an estimated uncertainty of 0.018 ppt (1-minute average; Brune et al., 2020)."**

Section 2.1 Page 3: As stated in the general comments, the analogous paper here is from Qu et al. ACP 2021 which uses V1.03 TROPOMI data whereas you use the Lorente et al. based corrections; make that difference clear here. Note that as far as I can tell there is fundamentally no difference between yours and the Qu et al. results, notwithstanding the improved XCH4 data sets you are using.

Can you add discussion on this difference in Section 2.1 and then add more comparisons to the Qu et al. 2021 results in Section 4?

**We have revised the text in section 2 to explain the differences between our study and that of Qu et al., as described earlier.**

**We have also modified the discussion in Section 4 to include comparisons with Qu et al. (2021):**

**"Eight of the ten nations in Table 2 (China, India, US, Russia, Brazil, European Union, Pakistan, Indonesia) are likewise identified by Worden et al. (2022) as among the top ten anthropogenic emitters globally. Our inferred anthropogenic fluxes for the US and China agree well (within ~10%) with the GOSAT-based results from Worden et al. (2022) and with the GOSAT+TROPOMI results from Qu et al. (2021). Anthropogenic emissions derived here are likewise within 10% of the Worden et al. (2022) results for India and the European Union, with both studies lower (20–50%) than Qu et al. (2021). Our results for Russia and Iran are 21–28% higher than the GOSAT-based estimates, mainly reflecting oil, gas, and coal emissions, and ~40% lower for Brazil, mainly due to livestock. Emissions for Pakistan and Indonesia agree to within 1% for the TROPOMI- and GOSAT-based results (Worden et al., 2022). However, we find here that anthropogenic emissions from Bangladesh (7 Tg/y versus a prior of 4 Tg/y) are 3× higher than the GOSAT estimate (2 Tg/y), while adjacent emissions from Myanmar (4 Tg/y) are half the GOSAT estimate. Worden et al. (2022) conclude that the GOSAT-derived emissions for Myanmar are anonymously high due to impacts from their prior assumptions; we attribute much of that flux to Bangladesh and show later that it mainly arises during the South Asian monsoon."**

**We have also added comparisons with GOSAT-based inversion results to section 5 and 6, as suggested by the reviewer in a later comment:**

**"Our derived global wetland fluxes are ~20% higher than previous GOSAT-based estimates (145-148 Tg/y: Ma et al., 2021; Zhang et al., 2021), with similar latitudinal distribution to that found by Ma et al. (2021)."**

**and**

**"The 2019 global anthropogenic methane emissions obtained here are modestly (12%) higher than GOSAT-based results for 2010–2018 (336 Tg/y; Zhang et al., 2021), with both results pointing to higher-than-predicted biotic emissions (consistent with isotopic constraints; Nisbet et al., 2016)."**

Section 2.6, page 7, Provide rationale for why you are downscaling to 0.1 degree resolution, especially since it depends on priors which can vary considerably (uncorrelated at 0.1 degree resolution) depending on choice of prior as you note in the text. As far as I can tell, the downscaled results are not used thereafter in the paper, is that correct? (Note that in the Worden et al. 2022 paper, we downscaled so that we can then upscale more accurately to each country; the other reason for the OE based downscaling (Cusworth et al. 2021) we developed is to step us towards using top-down emissions estimates for updating gridded inventories at this scale).

**The existing text already provided some rationale for this. We have now elaborated on that, adding the additional motivation mentioned by the reviewer regarding top-down satellite-informed emission inventories (as indicated below). The downscaled results are in fact used later in the paper to map the advance and retreat of East Asian emissions.**

Section 2: "We present here a new method to spatially downscale the satellite-derived emissions for potential use in models. This downscaling is necessitated by the fact that the current GEOS-Chem adjoint model does not have global simulation capability at finer than 2° × 2.5° resolution. Furthermore, each of the 2-year inversions performed here required >12,000 CPU hours (>80 days on multiple processors) to converge, making higher-resolution optimizations computationally impractical. However, the inventories employed as prior, as well as the TROPOMI observations themselves, contain information at much finer scales (e.g., 0.1° × 0.1° and 7 × 7 km2)—and thus contain additional high-resolution constraints that are neglected by the 2° × 2.5° inversions. We therefore leverage this information to spatially downscale the optimized emissions to 0.1° × 0.1° …"

As described earlier, we have added a caveat regarding the reliance on robust prior error estimates:

Section 2: "The method thus assumes robust prior error estimates, a caveat that also applies to Cusworth (2021) and similar methods."

How does this downscaling approach compare to the optimal estimation based approach to downscaling discussed in Cusworth et al Earth Environ 2, 242 (2021).  Can you perform a test(s) similar to what is shown in Cusworth et al. to ensure you are preserving information from original grid and downscaled grid? Your co-author A. Bloom designed these tests for Cusworth et al. so you could ask him for details. Note that I would be ecstatic for an additional vetting of this OE/Cusworth approach by the Dylan / Daven crew… we are pretty sure we got the math right as we used two different approaches to arrive at the same result (the Cusworth / Bloom and the Bowman approach, with Worden moderating), but given that its a 30+ equation derivation some additional vetting is desired.

Also cite Liu, M. et al. A New Divergence Method to Quantify Methane Emissions Using Observations of Sentinelâ5P TROPOMI. Geophys Res Lett 48, (2021), as a potential way to use satellite data to identify and quantify emissions at these same fine spatial scales.

This downscaling approach differs significantly from that of Cusworth et al. (2021) and we have added a discussion of this point to the paper as described earlier.

The test mentioned by the reviewer and presented in Cusworth et al. (2021) specifically evaluates the sectoral partitioning of that method. Our approach is purely spatial so such a test is not applicable. Instead we performed a dedicated OSSE experiment that demonstrates the performance and robustness of the method. This evaluation is described in the paper and we have now included the additional evaluation shown in (new) Figure S8 and described above.

Section 3.2. As a reader I did not understand either the rationale for the comparison to ATOM, or how I should interpret the comparison…. This section basically needs a re-write.  Note that our group at JPL also attempted to use the ATOM OH estimates but decided against it (although this was a few years ago) because we did not have a good sense of the accuracy, especially since OH is tricky to measure; some discussion is needed on the ATOM OH accuracy to better interpret the comparison between your inversion results and these in situ results. Also, what did you intend to conclude from the comparison to CO?

We have added new information to Section 2 regarding both the motivation for using the ATom data and the measurement uncertainties:

"We further use CO and OH measurements from the Atmospheric Tomography (ATom) airborne campaign (Wofsy et al., 2018) to test inversion success at separately optimizing methane sources and sinks. ATom featured pole-to-pole profiling (0.2 to 12 km) during four seasons over four years. The flight design is thus well-suited to determine whether the optimized OH fields improve or degrade global model simulations of OH itself and of CO (whose dominant sink is reaction with OH). Measurements of CO during ATom were performed using the NOAA Picarro instrument with an estimated uncertainty of 3.6 ppb (Chen et al., 2013). OH measurements during ATom employed the Airborne Tropospheric Hydrogen Oxides Sensor (ATHOS), with an estimated uncertainty of 0.018 ppt (1-minute average; Brune et al., 2020)."

We have also modified Section 3.2 to include additional statistical tests and to interpret the model-measurement differences in the context of measurement uncertainty:

"With the exception of ATom 3, the mean model OH biases with respect to ATom observations are ~80% lower for fixOH than for optOH (mean differences are all significant based on a paired t-test at 95% confidence). These optOH results exhibit a consistent OH underestimate (averaging 0.020–0.044 ppt) that exceeds the 0.018 ppt measurement uncertainty. Biases in the simulated background CO levels are likewise lower (by 7–87%) in the fixOH simulations, with a clear CO overestimate for optOH (Figure S9). Again, the mean fixOH/optOH differences are all statistically significant at the 95% confidence level, with model-measurement discrepancies for optOH (7–12 ppb) exceeding the 3.6 ppb measurement uncertainty."

Regarding the rationale for the ATom comparisons and the conclusions drawn from them, we believe these are made clear in the updated version from the following statements:

Section 2: "We further use CO and OH measurements from the Atmospheric Tomography (ATom) airborne campaign (Wofsy et al., 2018) to test inversion success at separately optimizing methane sources and sinks."

Section 3: "Remote observations of OH and CO (the primary OH sink) from the ATom airborne campaign (Wofsy et al., 2018) also point to an OH underestimate in the optOH solution."

Section 3: "When co-optimizing methane emissions and loss we thus find that the solutions can achieve a good fit to the TROPOMI data themselves but degrade model agreement with other observations of methane, OH and CO. We conclude that solving for global methane sources and sinks based solely on satellite observations of methane itself remains an under-constrained problem—even with the dense TROPOMI data coverage."

Section 5.0, Compare against the Ma et al. 2021 and Zhang et al. 2021 wetland results which suggest ~149 Tg CH4/yr total…this again might be a TROPOMI versus GOSAT issue as TROPOMI data results in lower livestock emissions than those from GOSAT in Brazil, which in turn would likely balance to the wetlands, relative to the GOSAT based results. A discussion here on these differences is needed.

Section 6, again compare these totals to the GOSAT based estimates (there are several now available). Discussion on potential TROPOMI / GOSAT differences are needed as well.

We have now added comparisons to GOSAT-based results as follows:

**Section 5: "Our derived global wetland fluxes are ~20% higher than previous GOSAT-based estimates (145-148 Tg/y: Ma et al., 2021; Zhang et al., 2021), with similar latitudinal distribution to that found by Ma et al. (2021)."**

**Section 6: "The 2019 global anthropogenic methane emissions obtained here are modestly (12%) higher than GOSAT-based results for 2010–2018 (336 Tg/y; Zhang et al., 2021), with both results pointing to higher-than-predicted biotic emissions (consistent with isotopic constraints; Nisbet et al., 2016)."**

**In addition, the paper already compared the country-level emissions with GOSAT-based estimates, as follows:**

**Section 4: "Eight of the ten nations in Table 2 (China, India, US, Russia, Brazil, European Union, Pakistan, Indonesia) are likewise identified by Worden et al. (2022) as among the top ten anthropogenic emitters globally. Our inferred anthropogenic fluxes for the US and China agree well (within ~10%) with the GOSAT-based results from Worden et al. (2022) and with the GOSAT+TROPOMI results from Qu et al. (2021). Anthropogenic emissions derived here are likewise within 10% of the Worden et al. (2022) results for India and the European Union, with both studies lower (20–50%) than Qu et al. (2021). Our results for Russia and Iran are 21–28% higher than the GOSAT-based estimates, mainly reflecting oil, gas, and coal emissions, and ~40% lower for Brazil, mainly due to livestock. Emissions for Pakistan and Indonesia agree to within 1% for the TROPOMI- and GOSAT-based results (Worden et al., 2022). However, we find here that anthropogenic emissions from Bangladesh (7 Tg/y versus a prior of 4 Tg/y) are 3× higher than the GOSAT estimate (2 Tg/y), while adjacent emissions from Myanmar (4 Tg/y) are half the GOSAT estimate. Worden et al. (2022) conclude that the GOSAT-derived emissions for Myanmar are anonymously high due to impacts from their prior assumptions; we attribute much of that flux to Bangladesh and show later that it mainly arises during the South Asian monsoon."**

Section 6.2, Note that reports to UNFCC from Russia have varied considerably over the years, this should be discussed here (e.g. Scarpelli et al. 2021 versus Scarpelli et al. 2022).

**We thank the reviewer for pointing this out and now discuss it in-text:**

**Section 6: "Indeed, subsequent revisions (year-2019; Scarpelli et al., 2022) to the UNFCCC-2016 inventory used here have strongly reduced fossil fuel emission estimates for Russia (e.g., 21 Tg/y from oil in year-2016 vs. 2 Tg/y in year-2019) due to updated emission factor assumptions."**

7.0 Conclusions (and to some extent abstract). The paper implies that missing sources can be identified through the downscaling approach, but this is not possible if you are using prior emissions for the downscaling. Also, how can the Venezuelan source simultaneously be lower than the prior and inline with trend estimates? These are different quantities. I think you mean something else here.

**The approach can in fact identify missing sources because it directly incorporates the downscaled TROPOMI observations themselves. As described in the paper, it is only the sectoral partitioning that relies solely on the prior, not the flux magnitude or location.**

**We have simplified the text about Venezuela for improved clarity:**

**Section 7: "We estimate long-standing fossil fuel sources in Venezuela at 4.8 (3.8–5.3) Tg/y, 28 (2–41)% higher than the prior estimate (which is for year-2016)."**

7.0: Line 540 Conclusions about waste and agriculture priors being too small… yes we are finding this to be the case with all the other published TROPOMI and GOSAT based inversions, please reference these other papers.

**We added comparisons to other relevant studies in the results and discussion sections, as described in our earlier replies.**

7.0 Conclusions / Line 555: This conclusion is potentially very interesting but needs additional vetting. For one, how much of yearly Indian and Southeast Asian underestimate is due to the underestimate in the Monsoon seasons? In addition, how much of this is affected by smoothing error, which is not directly calculated using your method, but you could calculate by using different priors; basically we are finding significant impact of smoothing error, or alternatively cross-correlation of a change in one emission onto another, for emissions and their trends in this region.

**Rather than using alternative priors, we have used a suite of inversions that employ substantially different optimization frameworks. These will be affected by smoothing to different degrees and we interpret and discuss our results in the context of the resulting uncertainty as diagnosed from the range across these solutions.**

**The reviewer's question about how much of the Indian/SE Asian underestimate falls during the monsoon is addressed by the following:**

**Abstract: "More than 45% of the global upward anthropogenic source adjustment occurs over India and southeast Asia during the summer monsoon (+8.5 Tg in Jul–Oct)."**

**Section 6: "Approximately 80% of India's annual rain falls during the summer monsoon (IPCC, 2021), and across this Jul-Oct season we find that methane emissions from the India and Southeast Asia boxes in Figure 2b are underestimated by 37 (15–45)%. The resulting seasonal flux increase then accounts for over 68% of the total annual emission correction."**

References: You can peruse the Worden et al. ACP paper for missing references on GOSAT inversions that you can then compare to in the text; this same comment was made by reviewers of our Worden et al. paper.

**We have revised the draft and cited these references accordingly, as indicated in our earlier relies.**

---

## Author Comment (AC2)

**Reply to community comments.**

**We thank the reviewers for the interest in our paper and the generous comments. We are glad that this paper has been useful for graduate coursework. Reviewer comments are provided below in black with our responses in blue.**

This review was prepared as part of graduate program course work at Wageningen University, and has been produced under supervision of dr. Ingrid Luijkx. The review has been posted because of its good quality, and likely usefulness to the authors and editor. This review was not solicited by the journal.

The paper by Yu et, al. entitled "A high-resolution satellite-based map of global methane emissions reveals missing wetland, fossil fuel and monsoon sources" presents a quantification of the 2018-2019 global methane budget, based on space-borne TROPOMI observations. Methane emissions are derived from the TROPOMI observations by coupling multiple 4D-Var adjoint inversions with a newly developed spatial downscaling approach. This enables the identification of previously missing or underestimated methane emissions from fossil fuel and wetland sources.

This research presents a new downscaling method that is applied to convert the GEOS-chem model output to a 0.1º × 0.1º resolution, using combined spatial information from the TROPOMI observations and from the prior estimates. This method enables very specific allocation of emission hotspots. In the study, OH is used as an additional constraint in the inversions, as recommended by Saunois et al. (2020). This advances previous studies, which often co-optimized methane sources and sinks by using methane data alone. The results section of the manuscript is well-written and discusses all the source areas in-depth, also suggesting possible underlying reasons for the found underestimations in prior inventories. The research reveals some interesting results regarding emission hotspots that were missing from prior inventories. That being said, I do have some remarks that could be addressed before publication.

**We thank the reviewer again for the positive comments.**

1) If I understand it correctly, the aim of the research is to quantify the 2018-2019 global methane budget and determine missing and underrepresented emission sources. However, the authors mainly present how much the prior estimates are underestimated compared to their findings, making the results section an evaluation of their one specific chosen set of prior inventories. This approach results in a high dependency of the research on the choice of the specific prior estimates. If other prior inventories were chosen, the underestimations and hotspots that the research now revealed would likely be very different, because for instance, as was pointed out in the introduction, two of the most commonly used anthropogenic emission inventories (EDGAR v5 and GEPA) are uncorrelated at a 0.1º x 0.1º resolution. To overcome this issue, I would suggest to shift the focus of the results section from the discrepancies in the specific prior to the obtained absolute values of the methane budget.

**We do include discussion of the changes relative to the prior inventories as these are widely used and their strengths and shortcomings are therefore of broad interest. However, we also already include discussion of the absolute flux amounts as the reviewer requests. For example, Table 2 and Section 4 list and discuss the derived sectoral fluxes both globally and by specific country. Section 5 then states the absolute wetland flux amounts for every region discussed (globe, Amazonia, central Africa, South Sudd, North America). Absolute flux amounts are also provided at relevant points throughout Section**

**6. In our view this provides a balanced discussion between the derived adjustments and the fluxes themselves.**

A good addition would then be a comparison of these results to independent measurements, such as the ObsPack or TCCON observations, or a comparison to other studies that also use inverse models to characterize the methane budget, such as Saunois et al. (2020).

**We already compare the results to ObsPack and TCCON observations, as follows:**

**"Ground-based methane column (XCH4) observations from the TCCON network (GGG2014 (2014)) show comparable improvements over the prior for both the fixOH and optOH solutions (and for their individual member inversions), with 71% (from -12.9 ppb to 3.8 ppb) and 66% (to 4.3 ppb) mean bias reductions, respectively (Figure 3, Table S1). However, global in-situ measurements from ObsPack (near-real time version v2.0; (2021)) reveal a 93% (from -13.8 ppb to -0.9 ppb) absolute mean bias improvement for the fixOH framework compared to just 39% (to 8.4 ppb) for optOH (Figure 3, Table S1). Figure 3 further shows that the optOH solutions overcorrect the prior negative bias with respect to ObsPack, providing a first piece of evidence for a methane sink underestimate in this inversion."**

**We have now updated our discussion of the derived national methane budgets to include comparisons with both Worden et al. (2021) and Qu et al. (2021):**

**"Eight of the ten nations in Table 2 (China, India, US, Russia, Brazil, European Union, Pakistan, Indonesia) are likewise identified by Worden et al. (2022) as among the top ten anthropogenic emitters globally. Our inferred anthropogenic fluxes for the US and China agree well (within ~10%) with the GOSAT-based results from Worden et al. (2022) and with the GOSAT+TROPOMI results from Qu et al. (2021). Anthropogenic emissions derived here are likewise within 10% of the Worden et al. (2022) results for India and the European Union, with both studies lower (20–50%) than Qu et al. (2021). Our results for Russia and Iran are 21–28% higher than the GOSAT-based estimates, mainly reflecting oil, gas, and coal emissions, and ~40% lower for Brazil, mainly due to livestock. Emissions for Pakistan and Indonesia agree to within 1% for the TROPOMI- and GOSAT-based results (Worden et al., 2022). However, we find here that anthropogenic emissions from Bangladesh (7 Tg/y versus a prior of 4 Tg/y) are 3× higher than the GOSAT estimate (2 Tg/y), while adjacent emissions from Myanmar (4 Tg/y) are half the GOSAT estimate. Worden et al. (2022) conclude that the GOSAT-derived emissions for Myanmar are anonymously high due to impacts from their prior assumptions; we attribute much of that flux to Bangladesh and show later that it mainly arises during the South Asian monsoon."**

**We have also added new comparisons to other prior top-down studies as suggested:**

**"Our derived global wetland fluxes are ~20% higher than previous GOSAT-based estimates (145-148 Tg/y: Ma et al., 2021; Zhang et al., 2021), with similar latitudinal distribution to that found by Ma et al. (2021)."**

**and**

**"The 2019 global anthropogenic methane emissions obtained here are modestly (12%) higher than GOSAT-based results for 2010–2018 (336 Tg/y; Zhang et al., 2021), with both results pointing to higher-than-predicted biotic emissions (consistent with isotopic constraints; Nisbet et al., 2016)."**

2) The authors take an ensemble mean of the 4 different inversion formalisms to calculate the emission corrections, while their previous research showed that some of them perform better for different purposes (Yu et al., 2021a). The different allocation of emission hotspots that are found through the different inversions are already nicely discussed in the results section, but the emission corrections are subsequently still calculated as the multi-model mean. I would like to see a more in-depth discussion of why the authors chose this approach, and why for instance the classical SF inversion is not left out here, since it is highly biased towards areas where the prior estimates of the emissions are high, and therefore likely makes the calculated underestimations of the prior estimates too small (Yu et al., 2021a). The BI inversion approach provides the best spatial distribution of all inverse approaches, while the EE inversion performs best in recovering large missing sources (Yu et al., 2021a). In the calculations of the hotspot emissions that are missing from the prior inventories, it is therefore probably better to use the EE inversion instead of the ensemble mean. Table S1 could also be used in this discussion, since it summarizes the performance of the different inversion formalisms, while the statistics presented here are currently not used in the text.

**We thank the reviewers for the interest in our previous work (Yu et al., 2021a). We chose to use the ensemble mean as our base-case solution as each individual inversion strategy has unique advantages. For example, in Yu (2021a) we found that the SF inversion exhibited the best performance for large sources already present in the prior, while the BG inversion yielded the highest spatial correlation with the true fluxes, and the EE inversion had the best performance for identifying missing sources. Furthermore, our previous OSSE was performed at ~25 km and the employed inventories had large spatial disparities at that scale. In this study, our inversions were performed at 2°x2.5° (~200 km) and the spatial inconsistencies are reduced at this larger scale.**

3) The section of the development of the novel downscaling method could be more extensive. Since a new method is presented here, it's very important that it is well-described. First of all, I would like to see the argumentation on why there was a need for a new downscaling method, and why previous downscaling methods were not suitable. Yu et al. (2021b) could be consulted, who present a nice review section of related work on spatial interpolation and downscaling of airborne pollutants.

**We have now added new text comparing our downscaling approach with other methodologies to better motivate its use, as suggested:**

**"Compared to existing emission downscaling methods that rely on prior and posterior error covariance estimates (Cusworth et al., 2021), or are based solely on satellite data (Liu et al., 2021), our approach is unique in combining the prior emission information (and its uncertainty) with the oversampled TROPOMI observations themselves. Variable weighting between these terms permits greater influence from the observations when the prior emissions are more uncertain. The method thus assumes robust prior error estimates, a caveat that also applies to Cusworth (2021) and similar methods."**

**Further motivation is provided earlier in Section 2.6, and this has been slightly expanded:**

**"We present here a new method to spatially downscale the satellite-derived emissions for potential use in models. This downscaling is necessitated by the fact that the current GEOS-Chem adjoint model does not have global simulation capability at finer than 2° × 2.5° resolution. Furthermore, each of the 2-year inversions performed here required >12,000 CPU hours (>80 days on multiple processors) to converge, making higher-resolution optimizations computationally impractical. However, the inventories employed as prior, as well as the TROPOMI observations themselves, contain information at much finer scales (e.g., 0.1° × 0.1° and 7 × 7 km$^2$)—and thus contain additional high-resolution constraints that are neglected by the 2° × 2.5° inversions. We therefore leverage this information to spatially downscale the optimized emissions to 0.1° × 0.1° via ..."**

Also, since the downscaling method is presented as novel, a proper evaluation of its accuracy is very important. I therefore wonder why the authors chose to perform the OSSE only for one area, for the duration of one month and at a resolution of 0.25º x 0.3125º, and subsequently chose to use a 0.1º x 0.1º resolution in their further research based on this OSSE. The representativeness of this one OSSE for the whole research should be better discussed and possibly expanded, since the validity of the research is dependent on this outcome.

**This OSSE evaluation approach was selected for several reasons. First, the GEOS-Chem model does not have global 4DVar optimization capability at 0.25°×0.3125° and does not have any optimization capacity at 0.1°. Even if it did, the computational cost would be far too high to run a global 4DVar evaluation at either scale. Second, the North American domain was selected as it includes all relevant source types and because we had simulation output available from Yu et al. (2021a).**

**To address the comment about accuracy we have now added a new downscaling bias reduction analysis, as follows:**

**"Figure S8 shows that the downscaled OSSE solution reduces the prior bias by 17%–56% for sources exceeding 1000 kg CH4/box/day (accounting for 99% of the domain-wide emissions) when not subject to transport error. In the presence of transport error, the downscaling method has limited success for the very largest sources (>2×105 kg/box/day), but nevertheless exhibits strong bias reduction (21%–50%) for sources between 1×103–2×105 kg/box/day (96% of domain-wide emissions)."**

[Figure]

**Figure S8. Downscaling bias reduction as a function of emission magnitude, based on 1-month Observing System Simulation Experiments (OSSE) over North America (see main text for details).**

4) After the results section, I would suggest to include a section where the uncertainties in both the TROPOMI data and the prior estimates is discussed, since the research is very dependent on both, and therefore also dependent on errors in the data. Also, the methods could be further discussed in this section, such as implications of the downscaling of the optimized emissions, and the use of the different inversion formalisms.

**Uncertainties in the TROPOMI data are already discussed in Section 2.1 based on both instrumental specifications and validation statistics versus GOSAT and TCCON observations. Our treatment of prior uncertainties is then discussed in detail in Section 2.4. We prefer not to duplicate that information in a separate section. The different inversion formalisms are already included in the discussion section based on the fact that we are using them to define our uncertainty range and to identify areas with consistent adjustment direction across inversions. The downscaled results are used later in the paper to map the advance and retreat of East Asian emissions.**

5) In my view, the knowledge gap could be further specified in the introduction. The novel aspect of the methods is already highlighted well by stressing the importance of including OH constraints, which many previous studies did not include. However, a section on prior knowledge about hotspots and emission sources that are often underrepresented in prior estimates is missing, including how the research is still of added value to this. Hu et al. (2018), who used TROPOMI to map methane column concentrations for instance also observed the underestimated hotspot of the Sudd wetlands and Venezuela. Lu et al. (2021) performed an inversion study using GOSAT data and also revealed missing spots in observational data, but on a far coarser resolution than this study. I suppose that the authors mainly add to this because of the far higher resolution of the TROPOMI data they use, combined with the downscaling method, making it easier to pinpoint emissions to more specific locations.

**Rather than expand the introduction (which would end up duplicating information provided later in the paper) we have instead expanded our comparisons to previous findings in the results section, as described above and in our replies to the other reviewers. We have added a citation of Hu et al. 2018 in the Sudd discussion as suggested.**

6) The authors nicely present the main underrepresented sources and missing hotspots in the conclusion, but a section with the further implications of these findings is missing.

**We believe that the implications are sufficiently covered in the results and conclusions and that an additional section is not necessary.**

 In the last section of the conclusion, some recommendations for future research are given (lines 561-564), but the statements include no references confirming that the addition of datasets of CO, methyl chloroform and formaldehyde would indeed improve future inversions. Also, the novel downscaling method is not mentioned in the conclusion, while this method is probably also relevant for further research.

**We have added some references supporting the use of CO, MCF, and HCHO, as follows:**

**"We address the issue in this work by employing validated OH fields from a chemical transport model, but future methane inversions can benefit from incorporating additional datasets (e.g., CO, methyl chloroform, formaldehyde) as constraints on the methane sink (McNorton et al., 2016; Rigby et al., 2017; Turner et al., 2017; Wolfe et al., 2019)."**

**The new downscaling methodology is in fact mentioned in the first paragraph of the conclusion section.**

Minor comments

Title: The current title is appealing because it directly mentions the new findings, but in my view, it does not cover the whole scope and innovative aspect of the research. I would consider changing the title to something like: "A high-resolution global map of methane emissions inferred from an inversion of TROPOMI satellite data reveals missing emission hotspots and previously underestimated sources."

**We thank the reviewer for this suggestion, but we prefer our current shorter title.**

Line 46: For a better overview of the previous research, I would elaborate here on what the conflicting reasons are for methane increase apart from the emission increase over tropical regions, such as an increase in emissions in the energy sector, an increase in wetland emissions, and a decrease in mean OH (McNorton et al., 2018).

**We thank the reviewer for this suggestion. In the introduction we are aiming to provide a robust overview of the state of science while still being concise. We believe the current text provides a suitable broad-level overview and does discuss the uncertainties arising from OH.**

Line 89: Please include the Sentinel-5 precursor/TROPOMI Level 2 Product User Manual Methane as a reference for requiring quality filter > 0.5: https://sentinel.esa.int/documents/247904/2474726/Sentinel-5P-Level-2-Product-User-Manual-Methane.pdf/1808f165-0486-4840-ac1d-06194238fa96

**We have added this citation as suggested.**

Line 96: Apart from mentioning the slope, please report the R2 as well here as a measure for agreement (R2 = 0.67).

**We have added these values as suggested.**

Line 117 - 128: Please elaborate on why these specific prior estimates are chosen, and perhaps also elaborate on how these datasets are constructed (by models/measurements)?

**We selected these inventories as they are commonly used and represent the current state-of-the-science. We refer the reader to the cited papers for more detailed descriptions.**

Line 118: Why did the authors chose to use the UNFCCC inventory from 2016? The new version from 2019 might be more representative for the study period.

**At the time we performed the inversion, that new version was not available. The global difference between these two versions is small for gas (24 Tg/y vs. 22 Tg/y) and coal (31 Tg/y vs. 33 Tg/y)**

**emissions, while oil sources decreased from 42 Tg/y to 26 Tg/y, mainly due to Russia. We have now added a discussion of this point to Section 6 as follows:**

**"Indeed, subsequent revisions (year-2019; Scarpelli et al., 2022) to the UNFCCC-2016 inventory used here have strongly reduced fossil fuel emission estimates for Russia (e.g., 21 Tg/y from oil in year-2016 vs. 2 Tg/y in year-2019) due to updated emission factor assumptions."**

Line 153: Please give a reference or explain why 50% uncertainty in the remaining sources is chosen.

**We have added a citation as requested.**

Line 162: It would be good to explain here how the OH sensitivity study is exactly performed, and specifically state where in the formula of the cost function the different uncertainties are used.

**We have clarified that the prior uncertainties for OH are included in Sa.**

Line 187: I wonder why the authors chose the values of 10% and 90% for the weight of the prior and the background respectively. Yu et al. (2021a) used 50% and 50% in their example of this background increment inversion formalism. Is this determined with sensitivity simulations similar as in the OG inversions? Please explain.

**Our previous OSSE was performed on a 25 km grid. At this resolution, the spatial distribution of prior emissions is highly uncertain. In this study, our inversions are performed at 2°x2.5° and the spatial distribution of prior emissions have higher fidelity, justifying the reduced weighting of the flat prior.**

Line 271: "Our 2019-2018 … growth rate acceleration": please elaborate on the implications of this statement on the findings that are presented in this paragraph.

**We have revised this text as suggested:**

**"Our 2018–2019 analysis timeframe also spans an El Niño, which has been tied both to global OH decreases and to methane growth rate acceleration (Anderson et al., 2021; Turner et al., 2018a)—further complicating a differentiation between the fixOH and optOH solutions."**

Line 314: Could the authors further explain here why the locations in the boxes of figure 2b were chosen for the analysis? This is probably because TROPOMI observations differ from the prior estimates in these areas. But when looking at the map, I see that this is for instance also the case for northern Italy and the Southeast US. Why are these areas not discussed?

**We selected areas with substantial methane emissions that cover a range of source types and reveal significant model-measurement discrepancies. In the interest of length, we are unable to examine every global region.**

Line 317: If I understand it correctly, the average yearly source and sink values for the years 2018-2019 that are presented here are not based on two full yearly cycles. The timeframe of the analysis only spans from 05/2018 - 10/2019. However, figure 4 indicates that the sources and sinks show seasonal variation. To retrieve yearly average values for the sources and sinks, these values can't be just averaged over a 1.5 yearly cycle. I would recommend to take these average values over one full yearly cycle, for instance from 10/2018-10/2019.

**Thank you for the suggestions. We have now clarified in the manuscript that "annual values discussed later are for 11/2018–04/2019 plus the average of 05–10/2018 and 05–10/2019."**

Line 412: I would move the explanation of figure 5c to line 396, since that is where the figure is first mentioned.

**We thank the reviewer for the suggestions. At line 396, we explain the emission corrections with comparison to previous studies. In line 412, we further discuss the underlying emission drivers. We think the current layout is more clear and have left it as-is.**

Line 442: Since these missing hotspots are one of the main outcomes of the research, the authors could consider to give their more exact locations, instead of only mentioning the countries.

**We have added an additional reference to Figure S16 for a clearer view of these locations.**

Line 447: I wonder how the hotspots can be missing in the UNFCCC inventory and show up in the EIA, since it seems like the UNFCCC is based on the national activity data from the EIA (Scarpelli et al., 2020). Is this because the authors used the UNFCCC data from 2016, and these activities were maybe still unknown at that time? Please explain this here, or as I mentioned before, consider using the updated UNFCCC inventory from 2019.

**The precise reasons that these sources would be missing from UNFCCC but present in EIA are not apparent to us and would need to be the subject of future work. As noted the updated UNFCCC version was not available at the time we performed our inversions, but we have added discussion of the UNFCCC updates to the manuscript.**

Figure 4: Please consider to make figure 4a-d larger, since the dots are very hard to see. Figure 4e is currently not referred to in the text. Also, I wonder why only the FixOH emission is shown here, and not the loss. I would either remove the fixOH emission from this plot, or include the loss as well.

**We have rotated Figure 4 to enlarge it and improve visibility as suggested. We have also added a reference to Figure 4e as requested. The fixOH loss is shown in Figure 4e since it is the same as the prior, this is now clarified in the figure caption.**

Figure 5: Figure 5a and 5c show information from previous research, while figure 5b shows main findings of the research. I would therefore suggest to make figure 5b a separate figure.

**Figures 5b and 5c are in fact both based on findings from this research, and we included 5a to provide context for interpretation.**

Figure S9: In my opinion, this figure could also be included in the main text, since it shows well how the outcomes of the four inversion formalism differ, and how the inversion ensemble is constructed.

**Thank you for the suggestion. In the interest of length, we choose to include Figure 4 in the main text and keep S9 in the SI.**

Specific comments

Line 17: Please remove "CO" here, since CO is not used as a constraint.

**We have revised this text as suggested. "Employing remote carbon monoxide (CO) and hydroxyl radical (OH) observations with independent methane measurements for evaluation, we infer from TROPOMI a global methane source of …"**

Line 43: "the importance of" can be left out here.

**We prefer to leave the phrasing as-is.**

Line 229: Write abbreviation of OSSE out in full.

**It is written out in full at first use, at the start of Section 2.4.**

Line 335: The total emissions of China mentioned here (60 Tg/y) is different from the number in table S2 (61 Tg/y). Please make this consistent.

**This has been corrected.**

Line 342: "Europe Union" > "European Union".

**Corrected.**

Table 2: "Russian" > "Russia".

**Corrected.**

References

Hu, H., Landgraf, J., Detmers, R., Borsdorff, T., Aan de Brugh, J., Aben, I., … & Hasekamp, O. (2018). Toward global mapping of methane with TROPOMI: First results and intersatellite comparison to GOSAT. Geophysical Research Letters, 45(8), 3682-3689.

Lu, X., Jacob, D. J., Zhang, Y., Maasakkers, J. D., Sulprizio, M. P., Shen, L., … & Ma, S. (2021). Global methane budget and trend, 2010–2017: complementarity of inverse analyses using in situ (GLOBALVIEWplus CH 4 ObsPack) and satellite (GOSAT) observations. Atmospheric Chemistry and Physics, 21(6), 4637-4657.

McNorton, J., Wilson, C., Gloor, M., Parker, R. J., Boesch, H., Feng, W., … & Chipperfield, M. P. (2018). Attribution of recent increases in atmospheric methane through 3-D inverse modelling. Atmospheric Chemistry and Physics, 18(24), 18149-18168.

Saunois, M., Stavert, A. R., Poulter, B., Bousquet, P., Canadell, J. G., Jackson, R. B., … & Zhuang, Q. (2020). The global methane budget 2000–2017. Earth system science data, 12(3), 1561-1623.

Scarpelli, T. R., Jacob, D. J., Maasakkers, J. D., Sulprizio, M. P., Sheng, J. X., Rose, K., … & Janssens-Maenhout, G. (2020). A global gridded (0.1× 0.1) inventory of methane emissions from oil, gas, and coal exploitation based on national reports to the United Nations Framework Convention on Climate Change. Earth System Science Data, 12(1), 563-575.

Yu, X., Millet, D. B., & Henze, D. K. (2021a). How well can inverse analyses of high-resolution satellite data resolve heterogeneous methane fluxes? Observing system simulation experiments with the GEOS-Chem adjoint model (v35). Geoscientific Model Development, 14(12), 7775-7793.

Yu, M., & Liu, Q. (2021b). Deep learning-based downscaling of tropospheric nitrogen dioxide using ground-level and satellite observations. Science of The Total Environment, 773, 145145.

---

## Author Comment (AC3)

**Reply to Review 2.**

We thank the reviewer for the thoughtful comments. Reviewer comments are provided below in black with our responses in blue.

The authors utilized the latest version of TROPOMI XCH4 retrievals averaged in 2018-2019 to optimize various sources of CH4 on a global scale. In addition, since OH and CH4 are intertwined, they added OH to the state vectors for adjustment (optOH). Finally, the authors proposed a statistical downscaling method leveraging both prior knowledge from bottom-up emission inventories and the oversampled TROPOMI data to scale the optimized 2x2.5 degree emissions down to 0.1x0.1 degree. This method enabled them to identify the missing sources better. Their primary take-home messages are i) The use of XCH4 observations is not adequate to provide reliable constraints on OH; this is why authors gave up on the optOH result; ii) the middle east underreports the energy-sector CH4 emissions; iii) In South and Southasia, there is a strong degree of correlation between CH4 agriculture and waste emissions and some hydrological variables such as more precipitation (or to be more precise, the runoff) during Monsoon seasons; iv) the reported emissions related to the oil and gas industry over the US in the latest bottom-emission inventory (EDGAR v6) is not too different than the top-down estimates made from this study. In general, the paper has important implications for the CH4 budget and regulations. However, the results are too optimistic because careful error quantification is lacking. In addition, some key aspects of inversions need to be clarified. A major revision is required to bring this draft to a publishable level.

We have addressed the reviewer's comments regarding uncertainty quantification and clarification; please see detailed responses to the reviewer's more specific comments below.

Major comments:

Inversion: While I understand the importance of using an adjoint model for implicitly resolving the source-receptor relationship without having to rerun the forward model multiple times, the inversion framework comes with a significant weakness which is its inability to gauge the confidence level in the final estimates (i.e., the posterior error). The paper should inform the readers about this major weakness (introduction, conclusion, and Table 2) and highlights studies such as Qu et al. 2021, which reported the AKs of the top-down estimates because they used an analytical inversion.

Our prior OSSE analyses (Yu et al., 2021a) showed that posterior error reductions calculated as the reviewer describes for methane do not in fact correlate with more accurate flux estimates when transport errors and spatial emission errors are present (as is generally the case). We have now added this information to Section 2:

"Adjoint 4D-Var inversions do not directly provide posterior error estimates. Methods are available to indirectly derive such estimates (Bousserez et al., 2015; Yu et al., 2021a). However, our previous Observing System Simulation Experiments (OSSEs; Yu et al., 2021a) showed that for methane the computed posterior error reductions do not correlate with more accurate flux estimates in the presence of model transport errors and spatial emission biases."

Because of this, we instead use an alternative strategy that combines multiple inversion frameworks and a wide suite of independent observations (for CH₄, CO, and OH) to evaluate our solutions and their uncertainty. This is now clarified in Section 2.4:

"Here, we instead combine multiple inversion frameworks (Section 2.5) with an ensemble of independent observations (Sections 2.1, 3) to test the robustness of our results and characterize the associated uncertainties."

And in Section 2.1:

"We use a large suite of independent measurements to evaluate the inversions. These include methane columns from TCCON (2014), a global network of Fourier transform spectrometers, and methane mole fractions from ObsPack (near-real time version v2.0; (2021)), a global compilation of ground-based and airborne measurements. We further use CO and OH measurements from the Atmospheric Tomography (ATom) airborne campaign (Wofsy et al., 2018) to test inversion success at separately optimizing methane sources and sinks. ATom featured pole-to-pole profiling (0.2 to 12 km) during four seasons over four years. The flight design is thus well-suited to determine whether the optimized OH fields improve or degrade global model simulations of OH itself and of CO (whose dominant sink is reaction with OH). Measurements of CO during ATom were performed using the NOAA Picarro instrument with an estimated uncertainty of 3.6 ppb (Chen et al., 2013). OH measurements during ATom employed the Airborne Tropospheric Hydrogen Oxides Sensor (ATHOS), with an estimated uncertainty of 0.018 ppt (1-minute average; Brune et al., 2020)."

To further address this comment we have now added a new OSSE-based evaluation of the flux accuracy achieved with our 4D-Var inversion + spatial downscaling approach, as described in our subsequent replies below.

Finally, we now highlight the differences between this study and that of Qu et al. (2021) as suggested:

Section 2: "Recent inverse analyses by Qu et al. (2021) likewise examined the global methane budget using TROPOMI (and GOSAT) observations. Our study advances on that work in several ways. First, we optimize monthly rather than annual fluxes to identify seasonal patterns of variability. Second, in place of a traditional analytical optimization we combine 4D-Var with new inverse formalisms for better identification of missing sources. Third, we develop a new downscaling approach to constrain emissions at high resolution, and use this framework to elucidate flux mechanisms and missing sources. Finally, our analysis leverages an updated TROPOMI methane product (Lorente et al., 2021) that corrects an albedo-dependent bias present in the version used by Qu et al. (2021)."

For example, can we trust the optimized CH4 emissions using the TROPOMI XCH4 over water or high SZA? are the reported top-down estimates statistically significant?

For the first point, we do not use retrievals over water or with high SZA (> 70°) and this is stated in Section 2.1 ("We omit high-latitude (>60°) observations and require quality filter QA > 0.5 (Sentinel-5 Precursor/TROPOMI Level 2 Product User Manual: Methane, 2022) to avoid errors associated with high solar or viewing zenith angles, low surface albedo, excessive aerosol loading, clouds, terrain roughness, and measurement noise (Lorente et al., 2021)") and Section 2.2 ("… over oceans (which lack TROPOMI XCH$_4$ data)..."). For the second point, as stated in-text we use the spread across diverse inversion frameworks to diagnose the level of confidence in the derived emissions. The associated uncertainty ranges are shown in Tables 2 and S2. Areas where derived emissions are not distinguishable from zero are also indicated visually as the shaded & hatched regions in the relevant figures (6, S12, S16).

In addition, several aspects of the inversion need to be further clarified: i) It needs to be explained how the 4D-var framework is applied when the inversion window is as wide as a 2-years average.

**As requested we have now clarified the inversion time window and temporal resolution in Section 2:**

**"Our inversions run continuously from 01/2018 to 02/2020, optimizing monthly grid-total methane emissions and 26-month mean hemispheric OH concentrations. To minimize any effects from initial conditions and to allow sufficient observational constraints throughout the analysis period we focus interpretation on the 18-month period from 05/2018 to 10/2019. Annual values discussed later are for 11/2018–04/2019 plus the mean of 05–10/2018 and 05–10/2019."**

ii) the TROPOMI full-physics algorithm relies on the prior profiles meaning the retrieved XCH4 is a piece of information on top of an ignorant model; I do not see any mention of if TROPOMI XCH4 was recalculated with GEOS-Chem prior profiles to ensure that only the true information from the satellite radiance is used for the inversion (Page 39 in http://www.tropomi.eu/sites/default/files/files/publicSentinel-5P-TROPOMI-ATBD-Methane-retrieval.pdf). This task should be done iteratively because the GEOS-Chem profiles change after each inversion iteration.

**Yes, this is stated in Section 2.2 and we now clarify that this is done at each inversion iteration:**

**"For all model-satellite comparisons (and at each inversion iteration) the GEOS-Chem output is sampled according to the TROPOMI observation operator at the overpass time and location."**

iii) it is unclear if the TROPOMI data have been scaled up to the resolution of GEOS-Chem in the inversion; if not, the difference in their spatial representativity will result in a perceived bias which can be problematic.

**We have now clarified this in the text. "For inversions on the 2° × 2.5° model grid, we first average the TROPOMI observations to this resolution."**

iv) how do the errors associated with the vertical diffusion in GEOS-Chem impact your result? The model error parameter is lacking in the analysis.

**As described in Section 2.4, our observing system error estimates are computed from the residual standard deviation between observations and prior simulations. This approach implicitly accounts for non-systematic measurement errors and model transport errors , and averages 11 ppb in our case. In separate ongoing work, we performed model simulations with an alternative vertical mixing scheme, and this leads to discrepancies that average 4.26 ppb. Explicitly accounting for such an effect would not appreciably change the error estimates employed here (e.g., (11^2 + 4.26^2)^0.5 = 11.8).**

v) is the state vector the total CH4 emissions, or is it sector-based?

**We optimize total emissions rather than on a sectoral level. We have now clarified this in multiple locations.**

vi) why did not the authors use the glint mode to account for off-shore emissions?

**Coverage locations for the glint mode off-shore observations vary from day to day and developing a framework to employ the sunglint data was beyond our scope.**

Downscaling: Two central problems exist: i) can the two-year average TROPOMI XCH4 truly capture the spatial variance in XCH4 at 0.1x0.1 degrees? By oversampling TROPOMI pixels, we may lose spatial variance (a smoothing effect) more than we reduce the random noises. The authors need to prove that a two-year averaged TROPOMI data can resolve the length scales of plumes at the resolution of 0.1x0.1 degree; if not, that resulting spatial representativity error induced by oversampling can potentially hinder reaching a 0.1x0.1 degree information.

**First, the TROPOMI nadir footprint is ~7 km, substantially smaller than the 0.1 degree grid size (~11 km) that we average to. Second, previous work has already demonstrated the ability of TROPOMI to resolve sources at far finer than 0.1 degrees. For example, Maasakkers et al. (2022, doi.org/10.1126/sciadv.abn9683) used other high-resolution satellites (GHGSat) and WRF simulations (at 3 km) to identify methane point sources, and showed that the TROPOMI methane measurements oversampled to 0.01 degrees (i.e., 10 times finer than employed in our work) can resolve these sources. The figure below, reproduced from Maasakkers et al. (2022) clearly demonstrates the TROPOMI ability to resolve features that are significantly smaller than 0.1 degrees. In our work we applied similar gridding methods but employ a coarser resolution of 0.1 degrees. We have added a citation of Maasakkers et al. (2022) to our paper.**

[Figure]

**Figure reproduced from Maasakkers et al. (2022, doi.org/10.1126/sciadv.abn9683). TROPOMI observations over Buenos Aires (Argentina). (A) Mean 2018–2019 TROPOMI methane concentrations oversampled (i.e., accounting for the full footprint of the observation) on a 0.01° grid. The Norte III landfill is indicated by the black cross; also shown are a GHGSat window centered on the TROPOMI-derived target (thick lines) and the Greater Buenos Aires municipalities [thin lines]. (B) A single TROPOMI overpass on 9 June 2019 exhibiting a methane plume downwind of Buenos Aires with wind arrows representing ERA5 10-m winds. (C) The 2018–2019 wind-rotated average giving a clear (north-oriented) plume signal indicating a concentrated source.**

ii) The proposed downscaling method (Eq2) heavily relies on assumptions about prior errors/information. Even the observational term depends on the prior fraction of emissions (fk). As noted by the authors, the prior CH4 emissions do not agree with other bottom-up emission inventories

(R2=0.01?), so how can one fully trust a downscaling output when it heavily relies on questionable prior information?

**The key point here is that for our approach incorrect prior emissions are acceptable provided that the errors are appropriate: when uncertainties are high, the observational term get weighted more heavily. Our spatial downscaling uses an error-based weighting term to balance between the prior information and the long-term TROPOMI data:**

**"As shown in Figure S7, the resulting downscaling relies most frequently on the prior information, particularly for low-emission areas, but hotspots and locations with higher prior uncertainties are preferentially informed by the observations."**

**We have now added a caveat regarding the reliance on robust prior errors, as suggested:**

**"The method thus assumes robust prior error estimates, a caveat that also applies to Cusworth (2021) and similar methods."**

**Third, the fk term does not impose any influence from the prior spatial distribution as the full $\sum_{k \in j} \left( f_k (y_{2y,k} - y_{bg,i}) \right)$ term in the denominator is simply a scalar that ensures conservation of the derived 2°x2.5° emissions.**

Would it be more sensible to use the posterior error/distribution for this part (under the condition in which the inversion framework was analytical, permitting the calculation of the posterior error)?

**As described in our earlier replies, our prior OSSE work (Yu et al., 2021a) has shown that larger error reduction does not in fact correlate with more accurate flux estimates.**

As a result of these two combined complications, I challenge the authors to provide an error estimation for this downscaling method and propagate them to the emissions maps and statistics (especially Table 2). Your study did not inform the posterior errors due to the use of adjoint; now, the lack of an uncertainty estimation for the downscaling part (which is the most crucial selling point of the paper) appears as an oversight.

**We have already addressed the comment about posterior errors and the issues with their use. The paper also already includes a dedicated OSSE analysis demonstrating the applicability and robustness of the downscaling approach presented here. To further address this point we now also include a downscaling error assessment based on the flux bias reduction achieved in the OSSE. This has been added to the text as follows:**

**"Figure S8 shows that the downscaled OSSE solution reduces the prior bias by 17%–56% for sources exceeding 1000 kg $CH_4$/box/day (accounting for 99% of the domain-wide emissions) when not subject to transport error. In the presence of transport error, the downscaling method has limited success for the very largest sources (>$2\times10^5$ kg/box/day), but nevertheless exhibits strong bias reduction (21%–50%) for sources between $1\times10^3$–$2\times10^5$ kg/box/day (96% of domain-wide emissions)."**

[Figure]

**Figure S8. Downscaling bias reduction as a function of emission magnitude, based on 1-month Observing System Simulation Experiments (OSSE) over North America (see main text for details).**

Comparison to ATOM: I need clarification on this comparison. Based on the author's discussion on optOH, they suggested that a strong El Nino year (2018-2019) led to lower-than-average OH mixing ratio (8.27e5 molec/cm3). But then they compared their constrained model in different years (<2018) with ATHOS OH measurements and concluded that their optOH is vastly underestimated, reinforced by the overestimation of CO. If 2018-2019 was a unique timeframe, how could one generalize the comparison results from other years to the 2018-2019 period?

**First, the model is sampled at the time of measurements in all cases, so that each comparison employs the same timeframe between model and observations. Second, the ATom comparisons span 2 years and show consistent patterns. Third, we also evaluated the inverse solutions using methane measurements for the inversion timeframe from TCCON and ObsPack. We arrive at consistent conclusions in all cases, lending support to our interpretation.**

Furthermore, I see a few issues here i) ATHOS OH can easily contain up to 30% error; have the authors considered the measurement errors in their comparison? Given the observational errors, I encourage applying a statistical test to know if the differences are real.

**First, we have added new information to Section 2 describing the ATom measurements and their uncertainties:**

**"We further use CO and OH measurements from the Atmospheric Tomography (ATom) airborne campaign (Wofsy et al., 2018) to test inversion success at separately optimizing methane sources and sinks. ATom featured pole-to-pole profiling (0.2 to 12 km) during four seasons over four years. The flight design is thus well-suited to determine whether the optimized OH fields improve or degrade global model simulations of OH itself and of CO (whose dominant sink is reaction with OH). Measurements of CO during ATom were performed using the NOAA Picarro instrument with an estimated uncertainty of 3.6 ppb (Chen et al., 2013). OH measurements during ATom employed the Airborne Tropospheric Hydrogen Oxides Sensor (ATHOS), with an estimated uncertainty of 0.018 ppt (1-minute average; Brune et al., 2020)."**

**Second, we now include additional statistical tests and interpret the model-measurement differences in the context of measurement uncertainty:**

**"With the exception of ATom 3, the mean model OH biases with respect to ATom observations are ~80% lower for fixOH than for optOH (mean differences are all significant based on a paired t-test at 95% confidence). These optOH results exhibit a consistent OH underestimate (averaging 0.020–0.044 ppt) that exceeds the 0.018 ppt measurement uncertainty. Biases in the simulated background CO levels are likewise lower (by 7–87%) in the fixOH simulations, with a clear CO overestimate for optOH (Figure S9). Again, the mean fixOH/optOH differences are all statistically significant at the 95% confidence level, with model-measurement discrepancies for optOH (7–12 ppb) exceeding the 3.6 ppb measurement uncertainty."**

ii) have the authors looked into the measured OHR to see if there are missing sources (such as VOC) in their model?

**This is out of scope as our model runs employ tagged methane and tagged CO simulations, which are offline and do not simulate other VOCs.**

What is the implication of underestimating OH in the optOH scenario? Should we perform a multispecies inversion with TROPOMI CO and HCHO to provide an additional constraint on OH?

**Our subsequent work is indeed pursuing analyses along these lines. We made the reviewer's point about the potential value of multi-species inversion at the end of the conclusions:**

**"We address the issue in this work by employing validated OH fields from a chemical transport model, but future methane inversions can benefit from incorporating additional datasets (e.g., CO, methyl chloroform, formaldehyde) as constraints on the methane sink (McNorton et al., 2016; Rigby et al., 2017; Turner et al., 2017; Wolfe et al., 2019)."**

In theory, letting OH see the CH4 feedback (optOH) is suitable, but why should the ATOM analysis discourage this meaningful practice? What if your default CO simulations are too high, and the optOH highlights that tendency? I'm left with many questions because the authors needed to dig into the problem more deeply.

**We arrived at this conclusion not just based on the ATom analysis but rather based on multiple consistent lines of evidence. First, independent evaluation of the posterior simulations against ObsPack observations reveals an optOH overestimate of methane concentrations, which would be consistent with an OH underestimate. Second, the ATom OH observations also point to an OptOH underestimate of OH. Third, the ATom CO observations reveal an optOH CO overestimate, which is also consistent with too-low OH. We agree with the reviewer that other factors can affect model-measurement agreement for CO but considered together these multiple consistent lines of evidence provide robust support for our interpretation as presented in the paper.**

Specific comments:

P1. L16. I do not think you ever used CO as an observational constraint for the model. This sentence is misleading.

**We have revised this text as suggested:**

*"Employing remote carbon monoxide (CO) and hydroxyl radical (OH) observations with independent methane measurements for evaluation, we infer from TROPOMI a global methane source of 587 Tg/y and sink of 571 Tg/y for our analysis period."*

P2. L38. Does really the recent enhancement in CH4 need to be better understood? I suggest adding more recent studies discussing the role of reduced NOx due to the lockdown on OH and CH4. There must be a recent study from Jacob's group regarding the increases in wetlands and permafrost CH4. This part needs more references in general.

**As suggested we have now added additional citations of recent studies by Stevenson et al. (2021) and Peng et al. (2022) discussing NOx emissions, wetland emissions, and their contributions to the recent methane increase.**

P2. L40. After reading the abstract saying that sinks and sources cannot be resolved with a high-resolution satellite, I found this sentence regarding transformative advancement somewhat contradictory.

**We have now revised this text:**

**"Employing remote carbon monoxide (CO) and hydroxyl radical (OH) observations with independent methane measurements for evaluation, we infer from TROPOMI a global methane source of 587 Tg/y and sink of 571 Tg/y for our analysis period."**

P2. L43. Shouldn't we also have an overrepresented source? If a source is underrepresented, another source should compensate for it.

**We have now revised this text: "missing and unexpected sources".**

P2. I found the second paragraph of the introduction imbalanced. The paper utilized remote sensing data, so I highly suggest comparing the pros and cons of using different remote sensing observations.

**We thank the reviewers for the suggestion. We prefer to keep this paragraph as-is, and compare our results with previous GOSAT-based inversions in the discussion section.**

P2. L60. Who came up with this R2 value? The agreement is unsettlingly low. Please provide a reference.

**We calculated this $R^2$ value directly from the cited inventories.**

P2. The third paragraph needs to include the temporal representation error between different emission inventories and the fact that CH4 emitters can vary from time to time at a relatively short temporal scale.

**We have revised this text as suggested:**

**"Meanwhile, current inventories also lack the ability to predict emission sporadicity (e.g., Irakulis-Loitxate et al., 2022; Pandey et al., 2019), while temporal representation errors can also arise between inventories due to time lags associated with their development."**

P3. L71. In the abstract, you said you had constrained CO, but here you imply that they will be used for evaluation.

We have now revised the abstract to clarify this point.

P5. L147. Why is the regularization factor applied to So instead of Se? We are less confident in Se compared to So. Another way (which should be the same as finding the maximum curvature in the L-curve) to find the optimum regularization factor is to scale Se several times and find the knee point in averaging kernels vs. the factor, although this might not be possible with the adjoint.

We apply the regularization factor to So following many previous studies (e.g., Jacob, et al., 2022, https://doi.org/10.5194/acp-22-9617-2022). We thank the reviewer for pointing out the knee point method.

P6. L167. How sure are you that the muted response of the model to a higher error in OH is not due to the lack of the degree of freedom? An analytical inversion would be able to answer it.

This is indeed one of our main conclusions: "evaluation of the inverse solutions indicates that methane sources and sinks cannot be simultaneously resolved by methane observations alone—even with the dense TROPOMI sampling coverage". In other words, the lack of degrees of freedom prevents robust simultaneous optimization of both sources and sinks.

Please see our earlier replies regarding analytical versus adjoint methodologies.

P6. L177. What is the implication of this low gamma value? The prior error is too uncertain or the observations are less noisy compared to the So?

The gamma value of 0.1 reflects the large number of observations compared to the size of the state vector.

P7. L213. Why should we accept this ad-hoc definition as XCH4 background? Any concerete evidence?

This XCH$_4$ background definition was determined via line search of the OSSE output. Specifically, we systematically varied the XCH$_4$ background parameters over a wide range and selected those that resulted in the best downscaling performance (i.e., giving the strongest consistency with the true local scale emissions). We have now revised the text to clarify this:

"This background definition was determined via OSSE analysis (described below). Specifically, the corresponding parameters were varied systematically over a wide range to identify values yielding the best consistency with the true underlying fine-scale emissions."

P9. Section 3.1. How can the errors in the soil uptake by methanotroph influence these results?

Soil uptake is only estimated to account for ~6% of the total methane sink, whereas oxidation by OH accounts for ~90% (Saunois et al., 2020). While uncertainties in other sinks can also exist, we therefore focus on OH here as it is the dominant sink and exerts the largest influence on methane source inversions. We have revised the text to clarify this point, as follows:

"Other minor sinks, such as soil uptake, are also uncertain but not addressed here."

P10. In the first paragraph, I encourage using absolute numbers from Figure 3 to describe the reduction in bias, such as (from -13.8 to 8.8 ppbv).

We have revised this text as suggested:

"Ground-based methane column (XCH$_4$) observations from the TCCON network (GGG2014 (2014)) show comparable improvements over the prior for both the fixOH and optOH solutions (and for their individual member inversions), with 71% (from -12.9 ppb to 3.8 ppb) and 66% (to 4.3 ppb) mean bias reductions, respectively (Figure 3, Table S1). However, global in-situ measurements from ObsPack (near-real time version v2.0; (2021)) reveal a 93% (from -13.8 ppb to -0.9 ppb) absolute mean bias improvement for the fixOH framework compared to just 39% (to 8.4 ppb) for optOH (Figure 3, Table S1). Figure 3 further shows that the optOH solutions overcorrect the prior negative bias with respect to ObsPack, providing a first piece of evidence for a methane sink underestimate in this inversion."

P10. L299-301. I'm afraid I have to disagree with saying that TROPOMI is dense, but we cannot fully resolve the source/sink of CH4. It would help if you had more than CH4 to get OH right (such as HCHO and CO constraints), which has nothing to do with the densityTROPOMI XCH4 observations.

Indeed, we agree with the reviewer and have addressed this point in our conclusions section:

"We address the issue in this work by employing validated OH fields from a chemical transport model, but future methane inversions can benefit from incorporating additional datasets (e.g., CO, methyl chloroform, formaldehyde) as constraints on the methane sink (McNorton et al., 2016; Rigby et al., 2017; Turner et al., 2017; Wolfe et al., 2019)."

P10. In the second paragraph, you should specifically mention what emissions are used for the retrospective simulations.

This information has now been added to Section 2.3:

"Simulations to evaluate posterior model performance for CO and OH employ anthropogenic emissions (for CO, NOx, and VOCs) from the Community Emissions Data System (Hoesly et al., 2018), the 2016 EPA NEI v1 (NEIC, 2019), and the Air Pollutant Emission Inventory (APEI, 2020). Corresponding biogenic and biomass burning emissions are obtained from the Model of Emissions of Gases and Aerosols from Nature (MEGANv2.1; Hu et al., 2015), and QFED (Koster et al., 2015)."

P11. Is rice cultivation part of the wetland?

No, we include rice cultivation in the anthropogenic category rather than in the wetland category. We have revised the text to clarify this.

"The TROPOMI-derived wetland fluxes (excluding rice) total 173 (155–182) Tg/y globally, representing 29 (26–31)% of the total methane source and 88 (84–91)% of the natural source."

P12. Why talk about livestock in the wetland sections?

Because here we are discussing uncertainties in the model partitioning between livestock and wetlands.

P12. 380. Where is the South Sudd in the figure?

It is shown as box 13 in Figure 2b and we have now clarified this in the text:

"Figure 2a shows that the South Sudd wetlands (box 13 in Figure 2b) are a major methane hotspot that is underestimated in the prior model by a column average of 41 (21–65) ppb."

P13. Can you provide more physical explanations of why these wetland emission models disagree so much? Is it due to their parameterization or the need for more information about water nitrogen content, heat content, depth of wetland, sulfate content, etc.?

**These wetland emission models are from the global carbon project and Saunois et al. (2020) describes their schemes as well as the associated discrepancies. Rather than repeat their discussion here we have added a citation as follows:**

**"Across the wetland regions examined above, Figure S14 shows that our optimized emissions fall towards the middle of the land-surface model estimates from the Global Carbon Project (GCP); details on these bottom-up models and their differences are provided by Saunois et al. (2020)."**

P14. The second paragraph: are we so clueless about the wetland anaerobic activity to use a simple correlation analysis? How about the soil nitrogen, water temperature, depth, oxygen content, etc.?

**While other factors can indeed control wetland emissions, our aim here is to set the stage and motivate future mechanistic studies similar to what the reviewer suggests. We now clarify this in the text.**

P14. L421. Please add a fraction of the total for each sector.

**We prefer to leave this sentence as-is, since adding all of the fractions impedes readability. The reader can readily calculate the fractions for themselves from the numbers provided in the sentence.**

P15. L444. Are they missing from other top-down emissions too? The bar is usually low for bottom-up emission inventories, especially in developing countries.

**As stated in the prior sentence we are referring here specifically to the inventories used as prior in our analysis.**

P17. L 515. Does correlation explain causation?

**Not necessarily and that is why we used the word "supportive" rather than something more definitive.**

P18. L 560. It's not about the density of TROPOMI data but a piece of factual information from XCH4. We need more compounds not denser data.

**We agree, please see our earlier replies regarding the potential utility of multi-species constraints.**

P16. is this number of available pixels really a lot? Please provide the percentage for a hypothetical situation when clouds were not present.

**We believe the absolute numbers stand on their own in this context.**

Editorial comments:

P1. L16. What do you mean by separately resolved?

**We have changed this wording to "simultaneously".**

P1. L20-21. It is vague; does the hydrological adjustment come after or before?

**We believe this wording is sufficiently clear as-is.**

P1. L22. The sentence (Fossil fuel emission...) is awkward.

**This sentence has been revised as requested.**

P1. L23. Many -> several

**We prefer to keep this wording as-is.**

P2. L39. What do you mean by strong heterogeneity? Spatial or temporal?

**We have clarified this sentence as requested.**

P2. L42. inverse -> inversion

**We have modified this sentence to "Here, we apply these data in a 4D-Var inversion + spatial downscaling framework to quantify the 2018–2019 global methane budget and determine the importance of missing and unexpected sources."**

P2. Please use +- for a normal range. 18+-1 is shorter and neater. Please apply this to the entire manuscript.

**The uncertainty range is not always symmetric about the mean and so we list the mean and range separately.**

P8. L242. What do you mean by "spatial source uncertainty"?

**We have revised the wording to "spatially biased emissions".**

P7. Eq.2. The i->j is weird; what do you mean?

**This describes the downscaling from coarse grid i to fine grid j.**

Figure 4. The panels are too small.

**We have modified the figure to improve visibility.**

Figure 5 needs to be enlarged. This is the most critical figure, which is hard to see.

**The figure has been enlarged as suggested.**

Table2. The numbers in the parenthesis are just the deviation in a defined box, not an actual error. Please inform the readers about it.

**We have added a footnote to the table to explain this, as suggested.**

---

## Author Response (AR2)

**Reply to Review 1**

We thank the reviewer for the constructive comments. Reviewer comments are provided below in black with our responses in blue.

General comments

The results in this version of the manuscript are better explained and should be published; however, there are many confusing word choices and comparison choices still present in the document. I therefore recommend that some of the more experienced authors in this manuscript carefully review this next version so that the next iteration is more likely to get through reviewer and editorial review.

In addition, as a reader, Im still struggling to restate the primary technical and scientific conclusions of the paper; I get that this is challenging as there are several (different types of) results described in this manuscript and summarized in the abstract…. Note that the short summary does a better job of summarizing results than the abstract so perhaps go over with your colleagues what you think the main results are and then come up with a succinct set of conclusions that can be stated in the abstract?

We have made some minor changes to the abstract for clarification and in response to other comments. After careful re-review we find that it is sufficiently clear and communicates our main findings well in its current form.

Specific comments

1a) Abstract, paper, and conclusion about estimating OH effect on methane emissions: I don't think you mean "cannot be simultaneously resolved" in the abstract but instead that you can estimate OH to XXX percent. For example, Im pretty sure you can falsify the hypothesis that the lifetime of CH4 (via OH) is greater than 100 year (or alternatively less than 1) using TROPOMI methane. Instead the paper simply states that you cannot easily choose between the fixed versus variable OH results so you went with the fixed because it agrees better with aircraft… this is not a bad argument but is very different from saying that TROPOMI alone cannot resolve the effect of OH on methane. Changes to the language are therefore needed to reflect what you actually did versus what you think is happening (error bars on the OH estimate would help here if you can provide them!)

We have revised the wording on this point in the abstract as requested.

1b) Note that the methane life time (or alternatively OH abundance) is also estimated in the cited manuscripts but there is no comparison to the estimated uncertainties in these manuscripts. As you do not report an uncertainty for the effect of OH on methane perhaps add some additional relevant discussion comparing your results to these other papers.

Instead of listing individual uncertainty estimates for every cited OH paper in Section 3, we give the range in derived global-average OH concentrations across those studies. This provides another uncertainty metric and makes the point that our two candidate OH fields both fall within a viable range based on prior studies of the global OH budget. Adding individual uncertainty estimates for the cited papers would not alter that conclusion.

2a) Abstract: Need error bars on results, e.g. 571 +/- ???, 20 +/- ???

Added as requested.

2b) Abstract line 25: don't use the word "global upward" and "India / Southeast Asia" in the same sentence as this is confusing.

**This sentence has been revised.**

3) Abstract and general text. Now that I better understand the downscaling approach , I concur that it is novel and should be published, However, as discussed in Liu et al. 2021 and other TROPOMI papers, variations in albedo can easily alias into the (downscaled) emissions. Some discussion is needed ion how you determine whether the downscaled emissions represent a real source versus a radiative artifact. Also, you could emphasize that these systematic errors are somewhat mitigated with the downscaling approach by weighting both the prior and conserving total emissions at the 2x2.5 flux estimate resolution.

**In our view this point is already addressed in the paper in the following sections:**

**"We use the SRON corrected retrieval described in Lorente et al. (2021), which is based on the S5P-RemoTeC full-physics algorithm with albedo correction and updated regularization scheme, spectroscopic information, and surface treatment. This updated algorithm mitigates the albedo bias that affected earlier versions (Qu et al., 2021). Relative to the albedo-corrected product, the prior TROPOMI version exhibits high biases over North Africa, the Middle East, and the western US, and low biases over Amazonia, the eastern US, central Africa, and eastern China (Lorente et al., 2021)."**

**"We omit high-latitude (>60°) observations and require quality filter QA > 0.5 (Sentinel-5 Precursor/TROPOMI Level 2 Product User Manual: Methane, 2022) to avoid errors associated with high solar or viewing zenith angles, low surface albedo, excessive aerosol loading, clouds, terrain roughness, and measurement noise (Lorente et al., 2021)."**

4) Section 2.6 Line 250: I suggest that the first sentence in this section adds some specificity to the downscaling approach, e.g. "We use a combination of the TROPOMI column enhancements and a priori to partition fluxes derived at 2.5x2 (lon/lat) degrees to 0.1 x 0.1 degrees resolution."

**This section has been revised accordingly.**

**"We present here a new method to spatially downscale the satellite-derived emissions from 2° × 2.5° to 0.1° × 0.1° for potential use in models. The downscaling, which combines information from the TROPOMI column enhancements, the prior emission estimates, and their uncertainties, is necessitated by the fact that the current GEOS-Chem adjoint model does not have global simulation capability at finer than 2° × 2.5° resolution."**

5) Abstract and Line 570: There are a couple of places in the text where you talk about the emissions magnitude but then you reference a trend… this type of comparison is very confusing.

This comparison approach is even more confusing in line 570 where you are saying a trend cannot fully explain the TROPOMI emissions. Compare mean to mean, not trend to mean.

**We thank the reviewer for pointing this out. In many cases our prior emissions are based by necessity on bottom-up inventories for previous years. A simple prior-to-posterior comparison therefore neglects any emission trend that is known or expected to have occurred between the time of the inventory and the time of our inversion. We therefore compare the derived emissions to the prior**

**value in the context of any trend that is estimated to have occurred in the interim. We have revised the text in multiple places for better clarity on this point.**

6) Line 576 units?

**Revised accordingly.**

7) conclusion on wetlands.. Is the 149 prior estimate for wetlands from Ma et al.? If so cite reference and note that this is informed from GOSAT data

**The 149 prior estimate is from Bloom et al. (2017) and Saunois et al. (2020). We have cited these references in section 2.3.**

8) Im still a bit confused about one of Monsoon conclusions. The results imply that not including the monsoon results in an underestimate of the emissions. However, is this underestimate relative to a yearly mean (i.e. yearly mean based on monthly emissions is higher than yearly mean that is generated by estimating yearly emissions total such as in the Qu et al. and Worden et al. papers). Or does it mean that emissions are underestimated during peak rainfall?

**The emissions are underestimated during peak rainfall, as follows:**

**"Approximately 80% of India's annual rain falls during the summer monsoon (IPCC, 2021), and across this Jul-Oct season we find that methane emissions from the India and Southeast Asia boxes in Figure 2b are underestimated by 37 (15–45)%."**

**Reply to Review 2**

We thank the reviewer for the constructive comments. Reviewer comments are provided below in black with our responses in blue.

The authors have added more caveats about their results and clarified ambiguous descriptions of their method. I wish the authors had put more time into answering my concern with the downscaler and analytically propagated their assumptions to the final products (for example, what if X and Y prior emissions are used?). I still think the robustness of the downscaler wasn't proven to me.

As explained in our earlier reply and in the manuscript, we have in fact carried out a thorough evaluation of the downscaling method. This includes dedicated Observing System Simulation Experiments (OSSEs) performed both with and without transport error. Results demonstrate the robustness of the downscaling approach both in terms of the spatial fidelity of the solution and in terms of the derived flux magnitude. Specifically, the downscaled solution "yields a larger bias reduction (98% versus just 16%) and more accurate flux distribution (R = 0.70 versus 0.60) than the native fine-grid 4D-Var solution". Furthermore, "the downscaled OSSE solution reduces the prior bias by 17%–56% for sources exceeding 1000 kg $CH_4$/box/day (accounting for 99% of the domain-wide emissions) when not subject to transport error". The OSSE results also demonstrate that "in the presence of transport error, the downscaling method has limited success for the very largest sources (>$2\times10^5$ kg/box/day), but nevertheless exhibits strong bias reduction (21%–50%) for sources between $1\times10^3$–$2\times10^5$ kg/box/day (96% of domain-wide emissions)."

Some minor comments:

"Spatial emission biases" > what do you mean by that? Do you mean the prior knowledge used as pseudo-observations in the Bayesian framework is uncertain?

We have clarified this wording.

"Here, we instead combine multiple inversion frameworks" > multiple inversion frameworks are vague. You did not perform a multi-species/multi-sensor inversion. Please clarify.

We have revised this wording.

Also, we cannot validate emission with observations alone because other underlying issues in the model exist that are not fully constrained; the most direct way of validating an inversion is to compare the posterior flux to flux observations (i.e., eddy covariance matrices).

We agree that observed fluxes enable a direct comparison, but such observations are extremely sparse and subject to significant representation errors when comparing to a global model.

ATom employed the Airborne Tropospheric Hydrogen Oxides Sensor (ATHOS), with an estimated uncertainty of 0.018 ppt > 18 ppqv

We prefer to keep these units in their present form.

In the presence of transport error, the downscaling method has limited success for the very largest sources (>2×105 kg/box/day), but nevertheless exhibits strong bias reduction (21%– 50%) for sources between 1×103 –2×105 kg/box/day (96% of domain-wide emissions)." > awkward grammar.

**Revised.**

Please cite Zhang and Jacob et al., 2021 who found correlated information in the posterior covariance matrices of emissions and methane lifetime, suggesting that their segregation isn't possible by using CH4 observations alone.

**We have added this citation.**

**A list of all relevant changes**

Line 16-32: word change and clarification

Line 58: add citation

Line 109-110: clarification

Line 116-119: clarification

Line 220-223: clarification

Line 234: clarification

Line 252-254: clarification

Line 271-272: clarification

Line 309: word change

Line 312-313: word change

Line 431: revise typo

Line 435-436: word change

Line 440-442: clarification

Line 481: word change

Line 541-542: word change

Line 568-570: clarification

Line 575-577: clarification

Line 587: word change

Line 594: word change

Line 608-611: clarification

Line 633: word change

Line 638-639: word change